# Efficient cavity-mediated energy transfer between photosynthetic light harvesting complexes from strong to weak coupling regime

Fan Wu [1], Tu C. Nguyen- Phan [2], Richard Cogdell [3] & Tönu Pullerits [1] ✉

Excitation energy transfer between photosynthetic light-harvesting complexes is vital for highly efficient primary photosynthesis. Controlling this process is the key for advancing the emerging artificial photosynthetic systems. Here, we experimentally demonstrate the enhanced excitation energy transfer between photosynthetic light-harvesting 2 complexes (LH2) mediated through the Fabry-Pérot optical microcavity. Using intensity-dependent pump-probe spectroscopy, we analyse the exciton-exciton annihilation (EEA) due to inter-LH2 energy transfer. Comparing EEA in LH2 within cavity samples and the bare LH2 films, we observe enhanced EEA in cavities indicating improved excitation energy transfer via coupling to a common cavity mode. Surprisingly, the effect remains even in the weak coupling regime. The enhancement is attributed to the additional connectivity between LH2s introduced by the resonant optical microcavity. Our results suggest that optical microcavities can be a strategic tool for modifying excitation energy transfer between molecular complexes, offering a promising approach towards efficient artificial light harvesting.

Excitation energy transfer is a photophysical process which forms foundation for many natural and man-made light-driven systems and devices. For example, in photosynthesis, excitation transfer between light harvesting complexes is the key primary step where the energy of the absorbed photons is driven to the reaction center where it is used for charge separation followed by a complex sequence of biochemical processes[1]. Being able to control or even optimize the energy transfer is of great significance to the development of artificial light harvesting systems[2–6]. In the past decade, strong light-matter interaction in microcavities, has been demonstrated to speed up the energy transfer process in various donor-acceptor systems[7–15]. Energy transfer between organic materials has been demonstrated even at a distance as large as 2 μm which is significantly greater than the Förster radius or the typical Frenkel-exciton diffusion length in such material[10]. Besides, a theoretical study has suggested that the energy transfer between photosynthetic light harvesting complexes can be enhanced 3 orders

of magnitude through the coupling with a cavity mode[16]. Recently, our group has reported the strong coupling between the light harvesting antenna 2 (LH2) complexes and an optical microcavity[17]. The coupling leads to the so called polaritons—the hybrid states of light and material excitations[18–20]. The corresponding LH2 polariton dynamics was significantly different from the excitation transfer in the bare LH2. The work is a step towards understanding how the cavity can control the excitation dynamics in light harvesting complexes. While our previous work reveals how the coupling with the cavity modes influences energy relaxation, we are not aware of any experimental study on the cavity-enhanced spatial energy transfer between photosynthetic antenna complexes, which will be discussed here.

To achieve strong exciton-photon coupling, one of the most widely used methods is to encapsulate the material in a Fabry Perot (FP) optical cavity[18–26]. The corresponding material states split into two new eigen energies, i.e., the upper polariton state (UP) and the lower

[1]Division of Chemical Physics and NanoLund, Lund University, Lund, Sweden. [2]School of Infection and Immunity, University of Glasgow, Glasgow, UK. [3]School of Molecular Biosciences, University of Glasgow, Glasgow, UK. ✉e-mail: tonu.pullerits@chemphys.lu.se

polariton state (LP). The energy separation between these two states is called the Rabi splitting $\hbar\Omega_R$, which is proportional to the effective coupling strength $g_{eff} = \hbar\Omega_R/2$ between the cavity mode and the material. The effective light-matter coupling strength depends on the number of molecules in the cavity mode volume $N$: $g_{eff} = g_0\sqrt{N}$[27,28]. Here, $g_0$ is the coupling between the transition dipole moment of an individual molecule and the field of the mode. Between the UP and LP states, there are $N-1$ states which form the so-called exciton reservoir with negligible contribution from the cavity mode and are therefore also called dark states. The effective light-matter interaction can be tuned by varying the concentration of the molecules inside the cavity. This changes the energies of the polariton states and the corresponding Rabi splitting. By tuning the energy levels, the relaxation processes[29–33], even chemical reactions[34,35], can be controlled. Here the influence of the dark states needs to be considered since the number of molecules can be very large, consequently the transfer between the polariton states and the exciton reservoir is important[36]. When reducing the concentration of the molecules, N goes down, the effective coupling becomes very small, and no Rabi splitting can be observed in a weak light-mater coupling regime. However, the coupling between the individual molecule and the cavity mode is still the same $g_0$. Can this interaction lead to cavity-mediated energy transfer even at the weak effective coupling regime? This question will be addressed in the current study by examining the exciton-exciton annihilation (EEA) between connected light harvesting complexes[37].

We will systematically investigate how the strong and weak exciton-photon coupling in FP microcavities influence the excitation energy transfer between photosynthetic light harvesting 2 complexes (LH2) from the purple bacterium *Rhodoblastus acidophilus*. The LH2 complex consists of two rings of bacteriochlorophyll (BChl) a molecules embedded into a protein scaffold. One ring is called B800 band, which contains nine BChl a molecules. The other is called B850 band, containing 18 closely packed BChl a molecules. Our recent work demonstrated strong and weak coupling between the B850 band of LH2 and FP microcavities[17]. In the current work we employ intensity-dependent pump probe spectroscopy to analyse the EEA due to the inter-LH2 energy transfer. Previously, the EEA has been analyzed in strongly coupled organic microcavity, and identified as a loss channel increasing the lasing threshold in polariton microcavities[38]. Here, we apply EEA to investigate the cavity-mediated inter-complex energy transfer in polaritonic systems for the first time. Comparing the EEA in strongly and weakly coupled LH2-containing cavity samples with the corresponding bare LH2 films, we uncover the effect of exciton-photon coupling on the excitation energy transfer between LH2s.

## Results and discussion

### Characterization of the strongly and weakly coupled LH2 containing microcavities

A schematic of the $\lambda/2$ FP microcavities used in this work is shown in Fig. 1a. The cavity consists of two 22 nm thick semi-transparent Au mirrors separated by a 300 nm thick layer of LH2 dispersed in a polyvinyl alcohol (PVA) matrix. Varying the concentration of LH2 in the PVA, three different microcavity samples containing LH2 were fabricated. Figure 1c shows the angular dispersion curve extracted from the typical angle-dependent absorption spectra of the high concentration LH2 containing cavity sample (see Supplementary Fig. 1). The

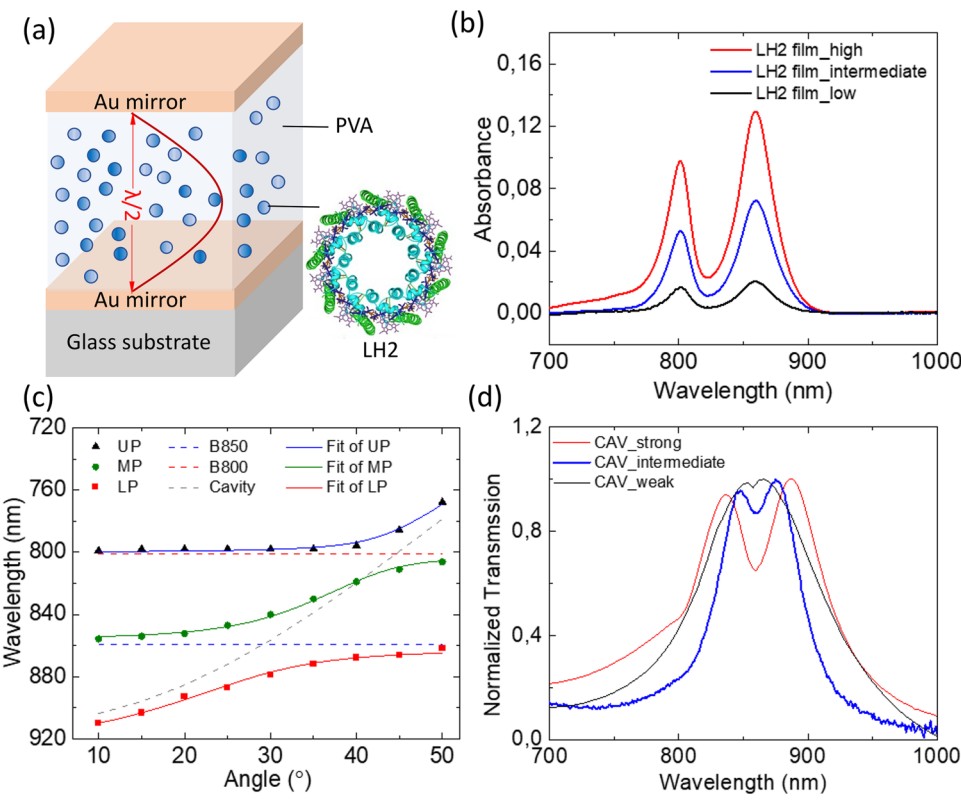

**Fig. 1 | Characterization of the strongly and weakly coupled LH2 containing microcavities. a** Structure of the semi-transparent $\lambda/2$ Fabry–Pérot cavity which consists of two semi-transparent Au mirrors (22 nm) enclosing a 300 nm thick PVA layer containing LH2; **b** steady-state absorption spectra of bare LH2 film on glass samples with well-resolved B800 band and B850 band of LH2, the high (intermediate, low) concentration LH2 film is prepared using the same spin coating solution as the strongly (intermediately, weakly) coupled LH2 cavity sample; **c** experimentally measured (scattered markers) and fitted (solid lines) angular dispersion curve of high concentration LH2 containing microcavity sample, where UP is the upper polariton branch, MP is the middle polariton branch and LP is the lower polariton branch; **d** steady-state transmission spectrum of high, intermediate and low concentration LH2 film containing microcavity samples, where the low concentration LH2 containing sample shows negligible splitting of the B850 band confirming weak light-mater interaction.

measured three polariton branches, i.e., upper polariton (UP), middle polariton (MP) and lower polariton (LP) are well fitted with the coupled oscillator model (see "Methods"). A clear anti-crossing behavior around the B850 band can be seen at around 30 degrees, demonstrating the strong coupling. A more detailed discussion in validating the strong coupling can be found in our previous study[17]. Decreasing the concentration of LH2 in the PVA, the obtained microcavity sample shows much smaller or even negligible peak splitting in their steady state transmission spectra, as shown in Fig. 1d, indicating that the B850 band of LH2 is intermediately or only weakly coupled to the cavity. According to the absorbance of the corresponding bare LH2 film on glass samples (see Fig. 1b), the average distance between LH2s was

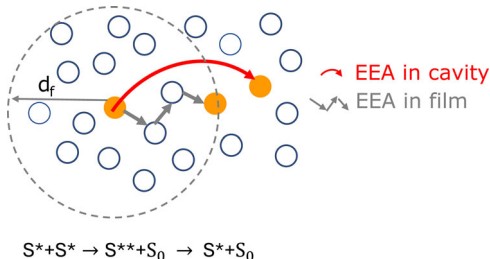

$$S^* + S^* \rightarrow S^{**} + S_0 \rightarrow S^* + S_0$$

**Fig. 2 | Schematic illustration of exciton-exciton annihilation (EEA) process.** The gray arrows indicate the EEA in bare film sample and the red arrow shows the EEA in the exciton-photon coupled cavity sample, where $d_f$ is the forster radius of the bare film, $S_0$, $S^*$, and $S^{**}$ is the ground state, first excited states, higher excited states of the chromophore, respectively.

estimated as described in Methods. For the high concentration LH2 film as well as the strongly coupled LH2 cavity, the average center-to-center distance of LH2s is around 9 nm. Considering the diameter of the individual LH2 (~7.6 nm)[39], the LH2s are quite closely packed in this case. In case of the low concentration LH2 film and weakly coupled LH2 cavity, the average distance is 16 nm and correspondingly the edge-to-edge distance between LH2s is 8.4 nm. We point out that the B850 BChl a molecules are embedded relatively deep inside the protein scaffold which means that the distance between the pigments of different LH2s is larger than the Förster radius of BChl a molecules of ~9 nm[40].

**Intensity-dependent femtosecond pump probe measurements**

To evaluate the excitation energy transfer between LH2s, we investigated EEA by using intensity-dependent pump probe spectroscopy. EEA is a process which involves the interaction of two excitons that can transfer energy within an array of molecules as illustrated in Fig. 2. As a consequence of the transfer, the two excitons can meet at a single molecule forming a double exciton—a higher excited molecular state which rapidly relaxes down to the lowest excited state, resulting in the annihilation of one exciton. Overall, the population of excited states decreases due to the EEA. With increased excitation intensity, the probability of EEA increases leading to a faster decay of the excited states and a shorter lifetime. Thus, excitation intensity dependent lifetime measurements are widely used to characterize the exciton transfer via EEA process. The pump wavelength was set at 785 nm, which excites the blue edge of the B800 band. Figure 3a shows the broadband pump probe spectra of the strongly coupled LH2 cavity

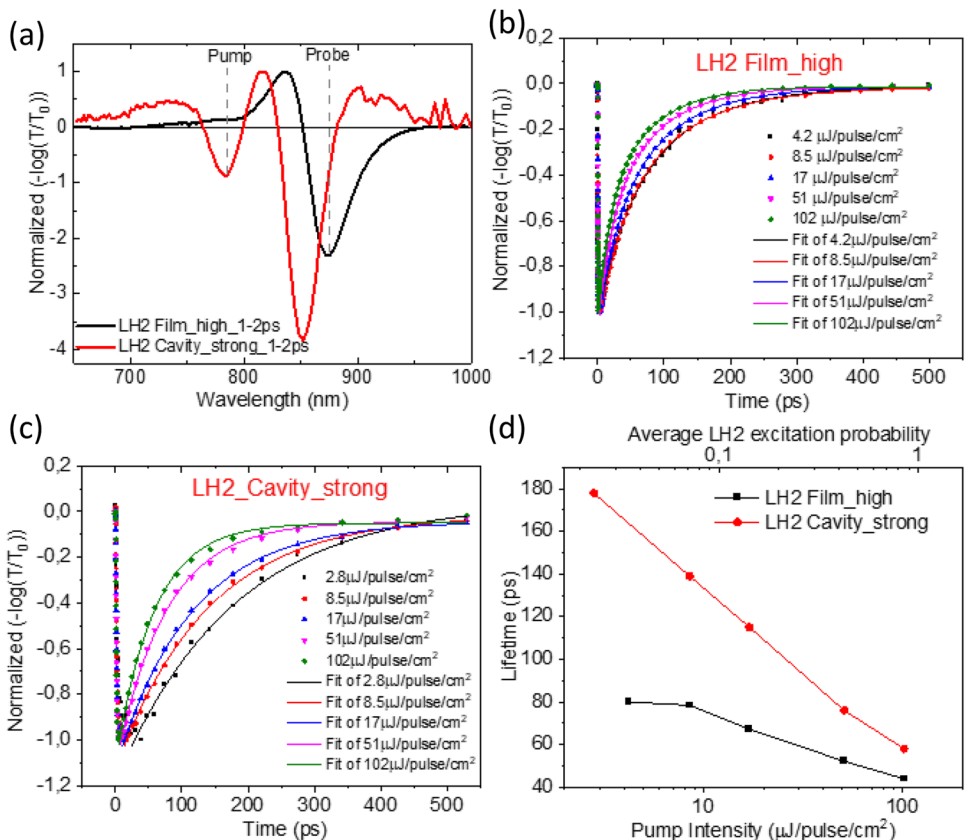

**Fig. 3 | Intensity dependent pump probe measurements of strongly coupled LH2 containing cavity sample and corresponding high concentration bare LH2 film sample. a** broadband pump probe spectra of strongly coupled LH2 containing cavity sample and corresponding high concentration bare LH2 film sample at 1–2 ps with pump at 785 nm; intensity dependent pump probe kinetics (scattered makers) at 875 nm of **b** high concentration bare LH2 film sample and **c** strongly coupled LH2 containing cavity sample; **d** the average lifetimes at different pump intensities obtained from exponential fit to experimental decays of strongly coupled LH2 containing cavity sample (red) and high concentration bare LH2 film sample (black). The solid lines are the exponential fits.

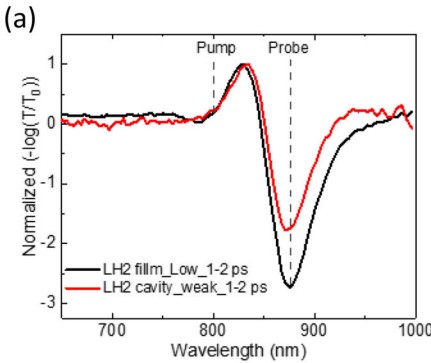

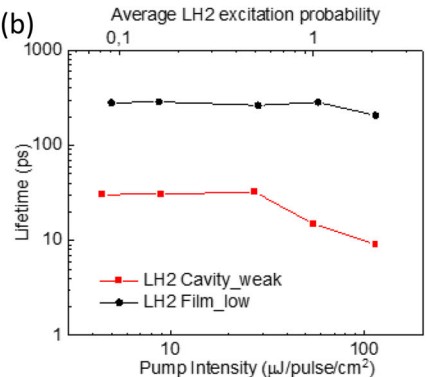

**Fig. 4 | Exciton-exciton annihilation results of weakly coupled LH2 containing cavity sample and corresponding low concentration bare LH2 film sample.**
**a** broadband pump probe spectra of weakly coupled LH2 containing cavity sample (red) and corresponding low concentration bare LH2 film sample (black) at 1–2 ps sample and the corresponding high concentration bare LH2 film sample which are averaged over 1–2 ps, respectively. The spectra at more delay times are displayed and discussed in Supplementary Fig. 2 and Note 1. The EEA processes of both samples were analyzed based on the kinetics at 875 nm at different pump intensities. Prior to any further analysis about the impact of strong exciton-photon coupling on the EEA process, it is important to exclude any non-polaritonic effects from the cavity structure, e.g., cavity induced light field change, which has been highlighted recently by Barnes and coworkers[41], where they examined cavity effects on the photoisomerization process between spiropyran and merocyanine, and found that the photoisomerization rates were correlated with the cavity induced absorption change. Here, we calculated the excitation intensity inside the cavity as compared with the excitation intensity outside the cavity which were employed for the bare film samples (see Supplementary Note 2). The results show that when the pump excitation wavelength is far off the cavity resonant energy, the pump intensities inside and outside the cavity are very similar, ruling out the above mentioned non-polaritonic effects. The signal at 875 nm corresponds to the GSB of B850 band in the bare LH2 films and GSB of LP in strongly coupled microcavity. Figure 3b shows the normalized intensity dependent pump probe kinetics of bare high-concentration LH2 film sample. All the kinetics are fitted with exponential decays. A more detailed analysis is shown in Supplementary Note 3. The lifetimes corresponding to the different pump intensities are plotted in Fig. 3d (black curve). When pump intensities are below 8.5 μJ/pulse/cm$^2$, the normalized pump probe kinetics are very similar, which implies conditions without EEA. Above the pump intensity of 8.5 μJ/pulse/cm$^2$, we can observe an obvious lifetime shortening with increased pump intensities, typical characteristic to an onset of EEA. At this pump intensity the average LH2 excitation probability is determined to be 0.07[17]. Here, we define the highest pump intensity before observing the lifetime shortening as the EEA threshold for the following discussion. Similar excitation intensity dependent pump probe kinetics were also measured for the strongly coupled LH2 containing cavity sample, as shown in Fig. 3c. In this case the pump probe kinetics are different even for the lowest two pump intensities, revealing an apparent EEA at as low excitation fluence as 2.8 μJ/pulse/cm$^2$ corresponding to an excitation density of 0.02 per LH2 ring or an excitation density of 0.07% per Bchl a molecule considering the number of Bchl a molecules per LH2 ring. Fitting the kinetics with exponential decays, we obtain pump intensity dependent lifetimes of the strongly coupled LH2 containing cavity sample as plotted in Fig. 3d (red curve). Comparing the lifetimes of the strongly coupled LH2 containing microcavity sample and the corresponding bare LH2 film sample at different pump intensities, we can see that the EEA threshold is much lower for the cavity sample, indicating an enhanced EEA, which also means an

with pump at 800 nm; **b** the average lifetimes at different pump intensities fitted with exponential decays of weakly coupled LH2 containing cavity sample (red) and low concentration bare LH2 film sample (black).

enhanced inter-complex excitation energy transfer mediated by the cavity mode. Here, we exclude that the EEA threshold could be affected by the untargeted effects that can be induced by the pump excitation, e.g., Rabi contraction, thermal expansion of the cavity, and bulk refractive index changes[42]. These non-polaritonic effects on the transient signals have been discussed and quantified in our previous report[17]. Supplementary Fig. 4 shows the calculated pump probe spectrum from these non-specific photoinduced effects based on the coupled oscillator model[17], which has a very different spectral shape and much weaker intensity compared to the experimentally measured spectrum. We point out that the strongest of these effects by large margin is the Rabi contraction which clearly depends on the excitation intensity. However, as reported by Musser and coworkers[42], the spectral shape of the transient signal that is related to the Rabi contraction is independent of excitation intensity and the signal strength scales linearly with the excitation intensity beyond 1% of the excitation concentration. This is much larger than the lowest two pump excitation densities where the lifetime shortening was still observed in this work. Therefore, we can exclude that the untargeted effects can influence the EEA threshold of the strongly coupled cavity sample. Polariton enhanced energy transfer has been earlier reported in other organic systems[13,43].

We applied the same investigation protocol for the cavities with significantly lower LH2 concentration corresponding to the weak coupling regime as demonstrated by the negligible Rabi splitting in Fig. 1d. Figure 3a shows broadband pump probe spectra of the weakly coupled LH2 cavity sample and the corresponding low concentration bare LH2 film sample which are averaged over 1–2 ps with pump at 800 nm. The spectra are very similar. The spectra at more delay times are presented and discussed in Supplementary Fig. 2 and Note 1. The corresponding intensity-dependent pump probe kinetics with probe at 875 nm are displayed in Supplementary Fig. 5. Fitting all the kinetics with exponential decays, the lifetimes corresponding to different intensities were plotted in Fig. 4b. The pump excitation threshold to initiate the EEA for the weakly coupled LH2 cavity is above 27.4 μJ/pulse/cm$^2$, which corresponds to an average LH2 excitation probability around 0.5, whereas the EEA onset for the corresponding low concentration LH2 film is above 57.7 μJ/pulse/cm$^2$, which means average LH2 excitation probability of around 1. A summary of the EEA threshold of the high concentration LH2 film, the strongly coupled LH2 cavity, low concentration LH2 film and weakly coupled LH2 cavity samples as discussed above is shown in Table 1. The lower pump excitation threshold for the EEA in the weakly coupled LH2 cavity sample as compared with that of the corresponding bare LH2 film shows that the cavity mode enhances inter-complex energy transfer even in the case of the weak coupling

regime. Since in the bare LH2 film, the energy transfer only exists between a limited number of nearest LH2s, the EEA can only be observed at high pump excitation, where almost all LH2s are excited. When LH2s are encapsulated inside a resonant cavity, every LH2 has the same average coupling $g_0$ to the cavity mode independently on the concentration of the sample. This induces additional connectivity between LH2s and makes EEA processes more efficient even in the case of weak coupling regime. The role of $g_0$ will be discussed further in the later Rhodamine 6 G (R6G) molecular system. The relevant differences between the strongly and weakly coupled cavity samples are the number of LH2s inside the cavity mode volume and the distance between LH2s. Comparing the EEA process in the strongly and weakly coupled LH2 cavity samples, we can see that the pump intensity threshold to initiate EEA is higher in the weakly coupled sample, indicating that the number of LH2s inside the cavity and the distance between LH2s play a role for the EEA efficiency. To confirm this conclusion, an intermediately coupled cavity sample was prepared, and the EEA process was studied as well. As shown in Supplementary Fig. 6, the EEA started at 8.2 μJ/pulse/cm², which is in between the thresholds for the strongly and weakly coupled cases. Also, the EEA process was less efficient in the low concentration LH2 film, compared to the high concentration LH2 film, since the distance between LH2s in the former case is much larger than the latter case, verifying the significance of the distance between molecules on the EEA process. Noteworthy, the lifetime of the weakly coupled LH2 cavity is shorter than the bare film as influenced by the short lifetime of the cavity mode (low Q-factor of the cavity). In contrast, the lifetime of the strongly coupled LH2 sample is longer than the corresponding high concentration bare LH2 film. This lifetime elongation by the strong exciton-photon coupling was addressed in our previous report[17], and is explained by the exciton reservoir model, where the excitation populates the so called dark states which have negligible contribution of the cavity mode and are formed due to the strong coupling. Since the number of dark states is immense (compared to only one short-lived lower polariton), their slow decay determines the long lifetime of the system.

To show the general applicability of optical microcavity to enhance the excitation energy transfer process between chromophores independently of the coupling regime, we also performed similar studies on a simple dye molecule Rhodamine 6G (R6G). The structure of the microcavity is as before. Here, Ag was selected as the mirror material because of its high reflectivity at the visible region where R6G absorbs. The active layer was prepared by spin coating a PVA solution which contains R6G molecules. Adjusting the concentration of R6G in the spin coating solution, we fabricated two different R6G containing cavity samples. Figure 5a shows the angle

**Table 1 | Summary of EEA threshold of the high concentration LH2 film, the strongly coupled LH2 cavity, low concentration LH2 film and weakly coupled LH2 cavity**

| Samples | EEA threshold | |
|---|---|---|
| | Pump excitation intensity (μJ/pulse/cm²) | LH2 excitation probability |
| LH2 film_high | 8.5 | 0.07 |
| LH2 cavity_strong | <2.8 | <0.02 |
| LH2 film_low | 54.7 | 1 |
| LH2 cavity_weak | 27.4 | 0.5 |

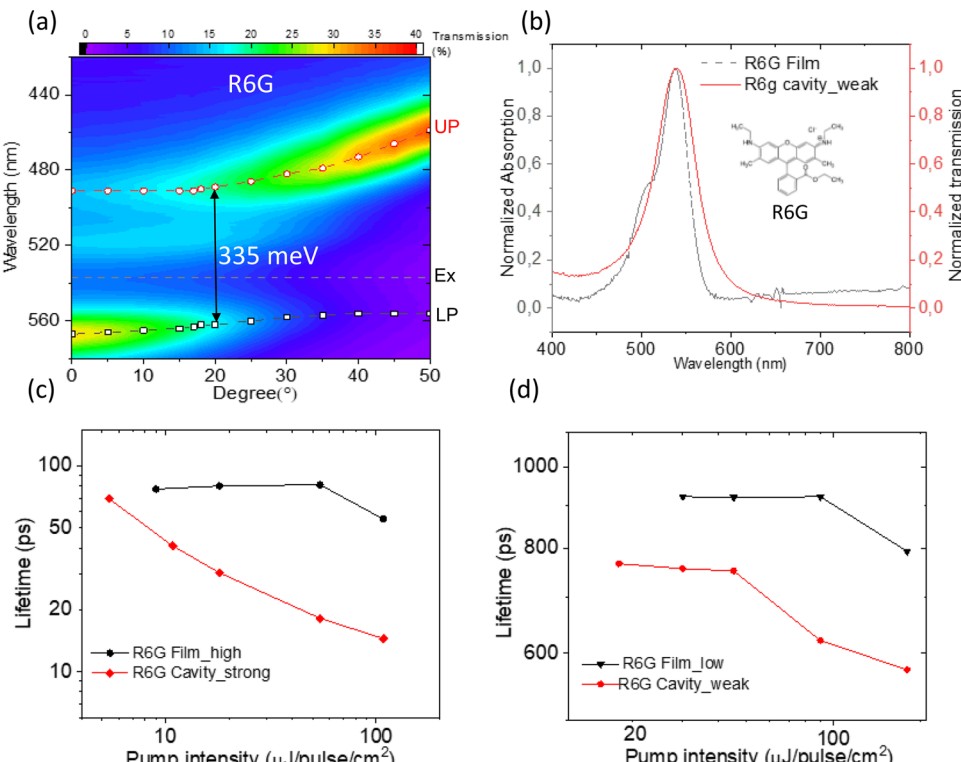

**Fig. 5 | Exciton-exciton annihilation study of strongly and weakly coupled R6G containing cavity samples and the corresponding bare film samples. a** Angular resolved (transverse electric mode) steady state transmission spectra of strongly coupled R6G containing microcavity sample; **b** steady-state transmission spectra (solid red line) of weakly coupled R6G containing microcavity sample and steady-state absorption spectrum of the corresponding low concentration bare R6G film on glass sample (dashed gray line); **c** the average lifetimes at different pump intensities fitted with exponential decays of strongly coupled R6G containing cavity sample (red) and bare R6G film sample (black); **d** the average lifetimes at different pump intensities fitted with exponential decays of weakly coupled R6G containing cavity sample (red) and bare R6G film sample (black).

dependent transmission spectra of the high concentration R6G containing cavity sample, where two new bands can be clearly resolved. The anti-crossing behavior of the two branches and the large energy splitting of 325 meV at 18° demonstrate that the system is in the strong coupling regime. When the concentration of R6G in the spin coating solution is decreased 10 times, the obtained microcavity sample shows no peak splitting in its steady state transmission spectrum as displayed in Fig. 5b, indicating the weak coupling regime. To evaluate the energy transfer between R6G molecules, analogous to the study on LH2, EEA between R6G molecules was examined using intensity-dependent pump probe spectroscopy. Supplementary Fig. 7 shows the pump probe kinetics of all the R6G containing samples with pump at 490 nm, and probe at 560 nm at different pump intensities.

Figure 5c compares the lifetimes obtained from the exponential fits of the pump probe kinetics of strongly coupled R6G cavity sample and the high concentration bare R6G film sample as a function of the pump excitation intensity. Obviously, even at the lowest pump excitation intensity, EEA was observed in the strongly coupled R6G containing cavity samples. While for the reference bare R6G film sample, there was no EEA at that pump intensity, unveiling an enhance EEA as well as energy transfer between R6G by the strong exciton-photon coupling. In addition, we also evaluated the relation of EEA on pump wavelength to generalize the conclusion of cavity enhanced EEA by pumping at 550 nm exciting the LP of the strongly coupled R6G cavity (see Supplementary Fig. 8). Noteworthy, when the pump excitation is near resonant with the cavity mode, the excitation intensity inside the cavity can be higher than the intensity outside the cavity as derived in Supplementary Note 2, which should be taken into consideration when evaluating the cavity induced effects. Here, the excitation intensity has been corrected accordingly by scaling the x-axis by the enhancement factor obtained from the calculation. Analogously, an enhanced EEA was detected for the strongly coupled R6G cavity, compared with the bare R6G film, indicating that the EEA is independent of the pump wavelength. The lifetime of the strongly coupled R6G cavity sample is shorter compared with the reference high concentration R6G film sample, which is opposite with the above LH2 case. The deviation was assumed to be related to the pump excitation intensity. Since the kinetics in the strongly coupled R6G shows EEA even at the lowest pump excitation, which means the intrinsic lifetime of strongly coupled R6G system may still be longer than the bare film sample. Figure 5d presents the lifetimes fitted from the exponential decays of the pump probe kinetics of weakly coupled R6G cavity sample and the low concentration bare R6G film sample as a function of the pump excitation intensity. Likewise, a lower EEA threshold was observed for the weakly coupled cavity sample in contrast with the low concentration bare R6G film sample, demonstrating an enhanced EEA, thus enhanced energy transfer process between R6G molecules by the weak light matter coupling. Here, we ascribed the exciton-photon coupling enhanced energy transfer to the individual exciton-photon coupling strength $g_0$, which brings additional connectivity between R6G molecules as compared to the corresponding film samples. To confirm the role of $g_0$ in affecting the EEA process, two more strongly coupled R6G cavity samples where the second cavity mode and third cavity mode are strongly coupled with the exciton energy, respectively (see Supplementary Fig. 9), were prepared through tuning the thickness of the middle R6G layer but keeping the concentration of R6G, i.e., the distance between R6G, the same in the cavity. In this case, the collective coupling strength $g_{eff}$ is the same for all the strongly coupled cavity samples, but $g_0$ is decreasing with increased thickness. Similar strong coupling of the high-order cavity mode with exciton energy has been reported for other dye molecules[44]. Furthermore, intensity dependent pump probe measurements were performed on these cavity samples with pump at 490 nm and probe at 560 nm. Analyzing the EEA results as shown in Supplementary Fig. 9, we can see that the threshold of the EEA is lower for thinner cavity. Therefore, we can

conclude that the larger the $g_0$, the lower the EEA threshold, demonstrating the more efficient cavity-enhanced energy transfer. Our results point at design strategies allowing to enhance the light-harvesting capacity of molecular systems by optical microcavities even within weak coupling regime. At the same time, the very efficient annihilation effect needs to be suppressed for the best results.

In conclusion, using excitation intensity dependent pump probe spectroscopy we have demonstrated an enhanced EEA between LH2s when they are strongly coupled to an FP microcavity, compared with the corresponding high concentration bare LH2 film. Surprisingly, the cavity-enhanced EEA was observed even in case of the weak coupling regime, revealing an enhanced energy transfer process as well. This strong and weak coupling enhanced energy transfer process is ascribed to the additional connectivity between molecules introduced by the resonant optical microcavities, which is contrasted to the uncoupled bare film samples, where the energy transfer takes place only between nearest molecules. The weak coupling enhanced EEA is less pronounced as compared with the enhancement in the strong coupling case, pointing out the role of the number of LH2s inside the cavity mode. The strong and weak coupling enhanced energy transfer also universally applies to simple R6G dye molecules. Our findings suggest that the resonant optical microcavities can be used as a design strategy to modify the excitation energy transfer process between molecules, and also a promising approach to optimize artificial photosynthetic devices. At the same time, care should be taken to reduce the adverse annihilation effect.

## Methods

### Sample preparation

LH2 complexes were isolated from *Rhodoblastus acidophilus10050* as reported previously[45] and dispersed in a TL buffer (0.1% LDAO, 20 mM Tris HCl pH 8.0) and stored at −80 °C as a stock solution. R6G was purchased from Sigma Aldrich. The strongly and weakly coupled LH2 containing cavity samples were prepared as described in an earlier work. And R6G containing cavities were prepared in a similar manner. Briefly, all the optical cavities were built on glass substrates (15 × 15 mm²), which were cleaned by successive sonication in alkaline solution (0.5% of Hellmanex in deionized water, 15 min), deionized water (15 min), acetone (15 min) and isopropanol (15 min), followed by an oxygen plasma treatment for 1 min. Subsequently, a semi-transparent metal mirror was deposited on the glass substrate by vacuum sputtering deposition (AJA Orion 5). The middle active layer was prepared by spin coating a polymer solution in which different concentrations of the chromophores were dispersed. Thereafter, a second metal mirror with the same thickness as the first mirror was deposited on top of the polymer layer also by vacuum sputtering deposition, which completes the cavity structure. As for LH2 containing cavity samples, the metal mirrors were 22 nm thick Au, and the middle active polymer layer was prepared by spin coating a mixed solution of LH2 with aqueous PVA solution. To prepare strongly coupled LH2 cavity and corresponding high-concentration LH2 film, the stock LH2 solution was mixed with a 8 wt % PVA solution at a ratio of 5:3. Oxygen scavengers were added to the mixed solution to protect LH2 from photooxidation[46,47]. For intermediately coupled and weakly coupled LH2 cavity sample, the concentration of LH2 is 4 times and 10 times diluted, respectively. The concentration of LH2 in PVA film indicated as high, intermediate and low is 2.5 mM, 1 mM, and 0.4 mM, respectively, which are calculated based on the Beer-Lambert law, knowing the absorbance of the films, the thickness of the film around 300 nm and the molar extinction coefficient of LH2[37] at 860 nm of $1.67 \times 10^6 \, M^{-1} cm^{-1}$. The distance between LH2s is calculated as[48]:

$$d = \frac{1.18}{\sqrt[3]{C}} \quad (1)$$

Where C is the concentration (mol/L) of LH2 in PVA film. The fabricated cavities were kept under vacuum in the dark at −20 °C to avoid any oxidation and any aging of the samples. For comparison, reference samples of bare LH2 films were fabricated using the same mixed solution and parameters for spin-coating on clean glass substrates without the Au mirrors.

In terms of R6G containing cavities, 30 nm thick Ag layers were sputtered as the semi-transparent mirrors. The middle active polymer layer was prepared by spin coating a 4% PVA aqueous solution containing 3 mg/mL R6G for the strongly coupled cavity sample or a 4% PVA aqueous solution containing 0.3 mg/mL R6G for the weakly coupled cavity sample on the Ag coated glass substrate. The cavity modes of all the cavity samples are tuned to be resonant with the R6G exciton energy.

### Steady-state spectroscopy

All the steady-state spectra were measured using a standard spectro-photometer (Lambda 1050, Perkin Elmer) with accessories. The angle-resolved transmission spectra were recorded having a variable angle accessory. A universal reflectance accessory was used to obtain the angle-resolved reflection spectra.

### Femtosecond pump probe spectroscopy

Ultrafast pump probe measurements were performed on two in-house built setups for single-color probe and broadband probe detections, respectively. The setups have been described in the previous report[17]. In brief, the broadband femtosecond pump probe measurements were carried out based on a Solstice (Spectra Physics) amplified laser system that produces ~60 fs pulses at a central wavelength of 796 nm at 4 kHz repetition rate. The laser output is split into two parts to generate pump and probe beams. The broadband white light probe was generated by focusing a 1350 nm pulse, which was produced by a collinear optical parametric amplifier (TOPAS-C, Light Conversion), on a CaF$_2$ crystal. To measure the ultrafast kinetics of LH2 containing samples (including the strongly and weakly coupled cavity samples and corresponding bare LH2 film samples), the pump wavelength was set to 785 nm with a pulse width around 100 fs, using a second TOPAS. For R6G containing samples, the pump wavelength was set to 490 nm or 550 nm using the same TOPAS. The pump intensity was varied by a neutral density filter. The single-color femtosecond pump probe measurement was employed to record the pump intensity dependent kinetics of the strongly coupled LH2 containing cavity sample and the corresponding high concentration LH2 film sample, using the following setup[49]. An amplified femtosecond laser system (Pharos, Light conversion) operating at 1030 nm and delivering pulses of 200 fs at a repetition rate of 1 kHz pumps two non-collinear optical parametric amplifiers (NOPAs, Orpheus-N, Light Conversion). One of them was used to generate pump pulses centered at 785 nm with a pulse duration of 100 fs. The second NOPA was used to generate probe pulses at 870 nm for differential transmission measurements. In both setups, pump and probe pulses were almost collinear. The probe was time delayed with respect to the pump by a mechanical delay stage. Polarization of the probe pulse was set to TE mode. The measurements were performed at room temperature. The data presented in this work are reported in terms of $-\log_{10}(T/T_O)$ to connect to the majority of transient absorption studies conducted outside cavities, where $T$ is the transmission measured with the pump pulse present and $T_O$ is the transmission without the pump pulse. The data were analyzed and fitted with exponential decays.

### Coupled oscillator model

The observed polariton branches in the strongly coupled LH2 containing cavity sample were fitted with a 3-by-3 coupled oscillator model described by the following Hamiltonian:

$$\begin{pmatrix} E_{Cav} & V_1 & V_2 \\ V_1 & Ex_{B850} & 0 \\ V_2 & 0 & Ex_{B800} \end{pmatrix} \begin{pmatrix} \alpha \\ \beta \\ \gamma \end{pmatrix} = E \begin{pmatrix} \alpha \\ \beta \\ \gamma \end{pmatrix} \quad (2)$$

where $\alpha$, $\beta$, and $\gamma$ are the mixing coefficients of the eigenvectors of the strongly coupled systems. The Hopfield coefficients in each polariton are given as $|\alpha|^2$, $|\beta|^2$, and $|\gamma|^2$. $E_{Cav}$ is the energy of cavity mode, and $Ex_{B850}$ and $Ex_{B800}$ the exciton energies of B850 band and B800 band. $E$ represents the eigenvalues corresponding to the energy of the formed polaritons. The cavity mode energy is given as[50] $E_{Cav}(\theta) = E_0 / \sqrt{1 - \left(\frac{\sin\theta}{n_{eff}}\right)^2}$. Here, $\theta$ is the angle of incidence and $E_O$ is the cavity energy at $\theta = 0°$. $n_{eff}$ is the effective refractive index. The coupling strength $V$ is related to the Rabi splitting $\hbar\Omega_R$, when $E_{Cav} = Ex$, it is given by $V = \hbar\Omega_R/2$. Here, $V_1$, $V_2$, and $n_{eff}$ are obtained and determined to be 24 meV, 31 meV and 1.48, respectively, by globally fitting the dispersion curves to the eigenstates of the Hamiltonian, using the Trust Region Reflective algorithm.

## Data availability

The main source data files supporting the findings of this study are provided in this paper. Any other relevant data is available from the corresponding author upon request. Source data are provided with this paper.

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

## Acknowledgements

F.W. and T.P. gratefully acknowledge Swedish Research Council (2021-05207), Swedish Energy Agency (44651-1 and 50709-1), Carl Trygger Foundation and NanoLund for funding this work.

## Author contributions

T.P. coordinated the project, F.W. fabricated the cavity samples and performed all the measurements, T.C.N.-P. and R.C. prepared the light harvesting complexes, T.P. and F.W. analyzed the data and formulated the conclusions. F.W. prepared the first draft of the manuscript. All authors contributed to discussion of the data and revision of the manuscript.

## Funding

## Competing interests

The authors declare no competing interests.
