## [Transparent Peer Review file · Nature Communications]

Efficient cavity-mediated energy transfer between photosynthetic light harvesting complexes from strong to weak coupling regime

Corresponding Author: Professor Tõnu Pullerits

Version 0:

Reviewer comments:

Reviewer #1

(Remarks to the Author)

Reviewer #2

(Remarks to the Author)

The authors present a comprehensive investigation of the cavity-mediated energy transfer process in LH2 complexes by performing intensity-dependent pump-probe spectroscopy experiments in a complete series of LH2 complexes in Fabry-Pérot cavities in the strong, medium, and weak coupling regimes together with the control systems with LH2 in bare films in the same concentrations as the ones used to prepare the cavity samples. In addition, in order to study the general applicability of optical microcavities to enhance energy transfer among chromophores, the authors present a similar study as performed for LH2 on a simple dye molecule (Rhodamine 6G). The results demonstrate an enhanced exciton-exciton annihilation between LH2 complexes, which in turn demonstrate enhanced excitation energy transfer between complexes, when they are within an optical microcavity even in the weak coupling regime. The authors ascribe the enhanced energy transfer to the greater connectivity among complexes due to the resonant cavity mode.

Overall, the manuscript is clearly written, the methods are properly described, and the conclusions are well-supported by the data presented. The work presented has wide-reaching implications since it presents a design strategy to optimize artificial photosynthetic devices, such as solar cells.

In summary, I consider that this work is in line with the journal's high quality multidisciplinary research standards. I recommend this manuscript for publication in Nature Communications. However, I do have some comments/questions that I would like the authors to address.

Comments

Page 4

"According to the absorbance of the corresponding bare LH2 film on glass samples (see Figure 1b), the average distance between LH2s was estimated. For the high concentration LH2 film as well as the strongly coupled LH2 cavity, the average centre-to-centre distance of LH2s is around 9 nm."

Could the authors explain in the Methods section how is this calculation performed?

Page 5

"Figure 2 (a) shows the broadband pump probe spectra of the strongly coupled LH2 cavity sample and the corresponding high concentration bare LH2 film sample at 1-2 ps, respectively."

What does "1-2 ps" mean? Does it refer to the average of the data collected between 1 and 2 ps? Please, clarify.

Typo: "The lifetimes corresponding to the different pump intensities are plotted in Figure 3c (black curve)." Fig3c should be Fig2d.

Page 7

I think that these two sentences: "Also, the EEA process was less efficient in the low concentration LH2 film, compared with the high concentration LH2 film, in terms of the pump intensity to initiate the EEA. Since the distance between LH2s in the former case is much larger than the latter case." should be merged.

Typo: In the sentence "Figure 4d presented the lifetimes fitted from the exponential decays . . ." The verb "presented" should be "presents" to be consistent with the rest of the text.

The sentence: "The middle active layer was prepared by spin coating a polymer solution which was dispersed different concentrations of the chromophores." Should not be something like: "The middle active layer was prepared by spin coating a polymer solution in which different concentrations of the chromophores were dispersed"?

Comments on figures

Fig1. To allow for better comparison among all LH2 samples studied and to better visualize the energy levels splitting induced by the coupling to the cavity mode, it will be good to display the absorption data for the three film samples in panel (b) and the three cavity samples in panel (c) which is currently shown in figS5.

Fig2a. Only the pump-probe data at 1-2 ps is shown (also in Fig3a). What is the reason for that? Could the authors show the pump-probe data for more delay times and discuss the spectral evolution of the different samples under investigation?

In Fig S5c I suggest to order the legend as in panel (a) for consistency.

Figures displaying absorption and transmission data: the use of the terms "absorption" and "transmission" in the figures seems to be misplaced in some of the graphs. For instance, in Fig1b "absorption" is used whereas in Fig1d "normalized transmission" is used; but both graphs show "absorption" data.

When comparing Fig4a and Fig2a in reference 17 (previous report from the authors) the graphs show equivalent data but in the current manuscript "Transmission (%)" is used while "Absorption (%)" is used in the previous report. Also, in Fig4b (current manuscript) two absorption spectra are displayed but, in the legend text, "transmission" is used for the cavity sample while "absorption" is used for the film sample, whereas in a similar graph in the previous report in Fig2b "absorption" is used for both cavity and film sample.

Obviously, an "absorption" spectrum is a "transmission" measurement but to use the term "transmission" and show "absorption" data is confusing and not correct. I suggest to always use "absorption" (as it is traditionally used in the photosynthesis research field and to ease comparison with the vast amount of data available for LH2 and as it is done with the pump-probe data displayed in units comparable with the majority of the studies conducted for LH2 outside cavities).

Reviewer #3

(Remarks to the Author)

The manuscript describes a time-resolved spectroscopy study of Light-Harvesting Complex 2 (LH2) containing PVA films in an optical cavity, in both the weak and strong light-matter coupling regimes. The strong light-matter coupling regime was characterized by the formation of polariton branches, which are identified by anticrossings of the cavity dispersion and the LH2 excitation maxima in the angle- and energy-resolved absorption spectrum.

By varying the intensity of the pump laser, the authors investigated how the cavity in the weak and strong coupling regimes affects the exciton-exciton annihilation (EEA) process in LH2, which is believed to involve the exchange of a photon between two excited bacteriochlorophylls. By comparing the excited-state lifetime of the LH2-cavity systems for various pump laser fluences, to the lifetime in the bare LH2 film, the authors find that the intensity threshold for EEA is significantly reduced in the cavities for both weak and strong coupling conditions. The authors interpret these findings as evidence for an enhanced energy transfer between the individual LH2 complexes in the cavities.

While enhanced energy transfer over distances exceeding the Foerster radius, has been observed under strong coupling conditions, we are not aware of reports of enhanced energy transfer also in the weak coupling limit. Also, from a theoretical perspective, enhanced energy transfer in the strong coupling regime seems (reasonably) well understood, but not for weak coupling. We therefore believe that the findings described in this manuscript will of high interest to a broad readership, and may motivate new research, both theory and experiment. We also share the author's opinion that the findings may have implications for artificial light-harvesting, as weak coupling is much easier to realize in practical applications, than strong coupling.

However, before we recommend publication, we would like the authors to clarify a few aspects, which are outlined below.

Our main concern is that the conclusions are based on measuring the lifetime at a single wavelength. The reason for our concern is that at the 875 nm probe, the difference spectrum, shown in Fig. 2a seems quite congested. Is it understood and does it matter, what transitions are probed: ground state bleach (GSB), stimulated emission (SE) or excited state absorption? Furthermore, is the extent of Rabi contraction intensity dependent? If so, this would only affect the evolution of the signal at 875 nm in the strongly coupled cavity, but not in the weakly coupled cavity or in the bare film. Focusing on the evolution of the transmission at a single wavelength may thus mask other processes. Therefore, would it be possible to acquire time-traces also for other wavelengths, in particular at 850 nm, where the difference signal seems strongest? Finding identical decay rates at multiple wavelengths would reinforce the conclusion that the increased decay rate at increasing intensity is indeed due to enhanced EEA.

The identification of the EEA thresholds seems based on a single data point in Fig. 2-4 (i.e. when these last two points are on a horizontal line). Leaving one point out could alter the conclusion, therefore. Would it be possible to acquire at least one more lifetime below threshold?

An interesting observation is that for the LH2 cavity systems, the slopes of the lifetime versus the logarithm of intensity are

different in the cavity and in the film. for the Rhodamine-6G (R6G) cavities, these slopes are identical. Do the authors have an explanation for this.

For the R6G system, the pump laser excites the lower polariton (LP), which is delocalized over many molecules, whereas for the LH2 system, the pump laser excites a chlorophyll in the B800 ring, which is a local excitation that then transfers into the polariton manifold that is formed due to the strong coupling of the B850 ring with the cavity mode. Because, as the authors point out, the relaxation pathway after excitation into the LP is different, we do not understand why a different excitation scheme was chosen. We believe that for a comparison between R6G and LH2, both systems should be excited off-resonantly into a dark state of the cavity-molecule system. This could, for example, be achieved if a higher lying electronic or vibronic state of R6G is pumped. We are not aware of any reports of such experiments, so perhaps this is not that easy. In that case, we suggest that authors discuss in more detail the implications of the different excitation conditions before generalizing their findings.

The authors base their conclusion of enhanced transport under weak and strong light-matter coupling conditions on a reduced excited-state lifetime. Because we think that this connection may not be immediately clear to the readers, we suggest to explain in more detail the EAA mechanism and the role of exciton transfer in that process. Perhaps a schematic illustration would also help.

The authors attribute the enhanced transfer of the excitation to the single molecule coupling (g_0), but also conclude that the number of molecules (or equivalently the number of dark states) plays an important role as well. Considering that the cavity with the highest concentration is just on the verge of strong coupling, also the cavity with the intermediate LH2 concentration must be in the weak coupling regime, yet the threshold is higher than in the cavity with the lowest concentration (Figure S5c). This would support the notion that the number of molecules is more important than the single molecule coupling. To understand the interplay between concentration, single molecule coupling and collective coupling better, would it be possible to vary the cavity thickness to be on resonance with a higher-order cavity mode, as in Bhuyan et al. (Adv. Opt. Mat. 12 (2024) 2301383)? This way, the single molecule cavity coupling can be reduced without affecting the collective coupling strength.

We have not understood the derivation in Supplementary Note 1, from which the authors conclude that the intensity inside the cavity is the same as outside. Intuitively, at every encounter with the mirrors there is loss due to transmission, while already at the first encounter of the incoming beam there is 70% loss due to reflectivity. This is very much like a transfer matrix (TMM) calculation, and we therefore wonder if a TMM calculations with suitable optical constants would give similar results?

In Figure S5c, one can see that for the lowest excitation powers, the lifetime increases with coupling strength (concentration). Are these differences due to the number of darkstates, or do these differences also depend on the overlap between the cavity spectrum and the molecular absorption spectrum, as suggested by Groenhof et al. in JPCL 10 (2019) 5476?

On Line 134, the authors refer to figure 3c, but according to us, this should be Figure 2d.

On Line 264: the authors state that a lower EEA threshold was observed for the weakly coupled R6G cavity than for the strongly coupled cavity, but from the data shown in figure 4, such threshold was not reached, as the cavity data points do not level off to a horizontal line in panels c and d.

Perhaps the authors can mention the wavelength in the text, on Line 131? In addition, we would consider it useful to also include the pump and probe wavelengths as lines in the spectra figures.

We have not understood how the authors computed the pump-probe spectrum from all untargeted effects (supplementary figure 3), in particular the Rabi contraction, which we believe depends on the intensity.

Reviewer #4

(Remarks to the Author)

Version 1:

Reviewer comments:

Reviewer #1

(Remarks to the Author)

in the revised manuscript, the authors have reasonably addressed the questions and concerns that I raised initially. I am happy with the revisions and therefore recommend this work for publication.

Reviewer #2

(Remarks to the Author)

I consider that the authors have made an effort to take into account all the reviewers' comments, they have made the necessary changes/improvements to the text and figures, performed new experiments, and added figures and text, which taken all together has significantly improved the manuscript.

Therefore, I recommend this manuscript for publication in Nature Communications in its current version.

Reviewer #3

(Remarks to the Author)

Because we used figures in our report, we uploaded it as a pdf file.

Reviewer #4

(Remarks to the Author)

Version 2:

Reviewer comments:

Reviewer #3

(Remarks to the Author)

Attached as pdf.

In addition we added an interactive Jupyter python notebook (eea.ipynb) with the commands we used to compute the absorption of R6G in the cavity. We included this file to help the authors sort out the discrepancies between our calculations and theirs, as suggested in our review.

Reviewer #4

(Remarks to the Author)

We thank you and the reviewers for the careful reading of the manuscript (NCOMMS-24-49180) and for the feedback that has helped us to significantly strengthen our work. We have thoroughly revised the manuscript and added new experimental results as requested by the reviewers. We offer answers to reviewers' comments and explain the changes in the manuscript. Besides, in the submitted revision, all changes are marked.

Sincerely, on behalf of the authors,

Tõnu Pullerits

REVIEWER COMMENTS

Reviewer #1 (Remarks to the Author):

This manuscript reports on cavity-mediated energy transfer in LH2 complexes. A hybrid between LH2 and Fabry-Perot cavity was studied by differential transmission in a pump-intensity-dependent manner to monitor exciton-exciton annihilation (EEA). Overall, this is a promising study. However, I have several reservations, listed below:

1) The authors emphasize that surprisingly, the effect is observed even in the weak coupling regime. I am not sure what is so surprising about it. And what is the interpretation of this result? Does this imply that the Purcell effect is the dominant mechanism at play?

Answer: Cavity enhanced energy transfer has mostly been reported in the strong coupling regime, but never in weak coupling regime. We point out that the strong coupling here is a collective effect and scales with \sqrt{N} , where N is the number of interacting molecules in the cavity mode volume. However, the intermolecular energy transfer is driven by the individual couplings between the molecules which in the first order does not depend on N . This means that even in the weak coupling regime the cavity enhances energy transfer by the additional connectivity induced by the coupling of the cavity mode with each LH2. This connectivity is present even when the number of LH2s is not large enough to enter the strong coupling regime. Since the Purcell factor of the Fabry-Perot cavity used in this work is low, the Purcell effect does not play a dominant role here.

2) On page 2, the authors use the symbol V_{eff} for collective coupling. This is highly confusing since this community's symbol " V " is typically reserved for the mode volume. Replace with g_{eff} or similar.

Answer: This has been changed.

3) For the main result of this work, displayed in Figures 1-2, it is not specified what is the concentration of LH2 in PVA, beyond low, high, and intermediate. The authors should be more quantitative. This is crucial.

Answer: The details have been added to the 'Methods'. The sentences of '*To prepare strongly coupled LH2 cavity and corresponding high-concentration LH2 film, the stock LH2 solution was mixed with a 8 wt % PVA solution at a ratio of 5:3.*' The concentration of LH2 in PVA film indicated as high, intermediate and low is 2.5 mM, 1 mM, and 0.4 mM, respectively, which are calculated based on the

Beer-Lambert law, knowing the absorbance of the films, the thickness of the film around 300 nm and the molar extinction coefficient of LH2³⁷ at 860 nm of $1.67 \cdot 10^6 \text{ M}^{-1} \text{ cm}^{-1}$ have been added.

4) The authors have chosen 785 nm for the pump and 875 nm for the probe. What motivates this choice? How will the results be affected if different choices of pump and probe wavelengths are made?

Answer: The pump wavelength was set to excite the B800 band of LH2 instead of direct excitation of the B850 band. This avoids the pump scattering in the probe region of the GSB of the B850 band. Also, in this excitation condition the light is far from the cavity resonance and therefore as explained in the Supplementary Note 2, the pump intensity inside the cavity and outside the cavity for bare film sample are the same. The probe wavelength corresponded to the GSB of lower polariton state of the B850 band for the strongly coupled LH2 cavity sample and the GSB of B850 band for bare LH2 films and weakly coupled LH2 cavity sample. This is an ideal choice of the probe for investigating the exciton-exciton annihilation between the B850 excitons.

Changing pump wavelength can influence what is initially excited. Therefore the details of the initial fast process where the energy is transferred to B850 band or the corresponding polaritons can differ. However, the arrival of the excitation to the B850-related states will be faster than 1ps anyway. Considering that the EEA between LH2s occurs at a time scale longer than tens of picoseconds, different excitations will not affect the EEA results as long as the excitation concentration is the same. In addition, to address this question we performed new intensity dependent pump probe measurements on the strongly coupled R6G cavity by exciting the UP with the pump wavelength around 490nm and probe around 560 nm as a supplementary result (see supplementary Figure 8) of the one already shown in the manuscript where LP was excited. The EEA was analysed and compared between the strongly coupled cavity and the bare R6G film. Likewise, the EEA threshold is still lower for the cavity sample than the bare R6G film, indicating the EEA process is independent of the pump wavelength.

The following text was added to page 13 of the manuscript: *'In addition, we also evaluated the relation of EEA on pump wavelength to generalize the conclusion of cavity enhanced EEA by pumping at 490nm exciting the UP of the strongly coupled R6G cavity (see Supplementary Figure 8). Analogously, an enhanced EEA was detected for the strongly coupled R6G cavity, compared with the bare R6G film, indicating that the EEA is independent of the pump wavelength.'*

As for different probe wavelengths, we compared the pump intensity dependent kinetics at 850nm and the kinetics at 875nm of the strongly coupled LH2 cavity sample. As shown in the Figure below, we can see that the lifetimes at the two wavelengths are the same, while the initial rising time at 850nm is a bit faster than the one at 875nm. Since the two states share the same ground states, it is reasonable to observe the same lifetime, and thus a similar EEA at these two wavelengths. While the difference at the initial rising time between 850nm and 875nm corresponds to the energy relaxation pathway from MP to DS and DS to LP, respectively.

Comment Fig1 Comparison of (a) the kinetics at 875nm and 850nm of the strongly coupled LH2 cavity sample at different pump intensities and (b) the initial rising kinetics at 875nm and 850nm. 5) Technical comment: In line 134, the authors write "Figure 3c", but they probably mean "Figure 2d". Check this.

Answer: This has been corrected. We thank the referee.

6) General comment: The most important results of this work are presented in Figures 1 and 2. The authors show that lifetimes are shortened both inside and outside of the cavity at high pump intensities. In the case of a strongly coupled system, the lifetime is, however, substantially longer than that of the uncoupled film. This lifetime elongation seems to be one of the most interesting results of this work. Yet, the authors claim (in lines 204-205) that this lifetime elongation was already discussed in Ref. 17. In this regard, in my opinion, the authors should emphasize what is different in this work in comparison to their previous work. Their message at the moment is not clear, which makes the publication of this work questionable.

Answer: The referee is right, one of the key observations of our earlier work was modification of the excitation dynamics by the cavity. For example, due to the dark states the lifetime prolongation was observed in the strong coupling cavity case. Those studies were carried out at low excitation intensities where the dynamics was independent of the excitation fluence. Contrary to that, here we investigate the excitation intensity dependence of the dynamics. This is the key difference between the two studies. The main point of the Figure 1 and 3 in the revised manuscript is the different onset of the lifetime shortening as a function of the excitation intensity. The lifetime shortening is reporting about the exciton-exciton annihilation connected with the energy transfer process between LH2s. This is very different from the exciton decay process in an LH2 studied in the previously published paper. Another difference is that the previous paper focused on how the strong exciton-photon coupling affect the dynamic process, while the current manuscript studied the effects on energy transfer between LH2s not only in the strong coupling regime, but also in the weak coupling regime, which has never been studied before.

In order to emphasise the differences of the two studies we have reformulated the text in page 2 as *'While our previous work reveals how the coupling with the cavity modes influences energy*

relaxation, we are not aware of any experimental study on the cavity-enhanced spatial energy transfer between photosynthetic antenna complexes, which will be discussed here.'

7) The authors also observe that the EEA threshold for a strongly coupled system is much lower than that of the bare LH2 film. How is this threshold determined technically? What are the values of these thresholds for each case, strong, weak, bare, etc. (perhaps provide a comparison Table)? Finally and most importantly, what is the mechanism of affecting the EEA threshold by strong/weak coupling? These issues must be clarified to warrant publication, otherwise the value of this contribution is not obvious.

Answer: The threshold is the highest pump intensity before we observe the lifetime shortening in the intensity-dependent pump probe measurements. The sentence of '*Here, we define the highest pump intensity before observing the lifetime shortening as the EEA threshold for the following discussion.*' has been added to page 7 of the main manuscript.

Table 1 has been added. The sentence '*A summary of the EEA threshold of the high concentration LH2 film, the strongly coupled LH2 cavity, low concentration LH2 film and weakly coupled LH2 cavity samples as discussed above is shown in Table 1.*' has been added accordingly to page 9.

Mechanism: We propose that it is the additional connectivity induced by the coupling of cavity mode with each exciton, i.e. g_0 , that makes the EEA process more efficient in both strongly and weakly coupled cavity samples, compared with the EEA in the corresponding bare film samples. The g_0 of the strongly and weakly coupled cavity samples studied in this work is the same, since all the molecules are distributed evenly in the cavity. Although we still observed different EEA threshold for the strongly and weakly coupled cavity samples which means that the number N also plays a role for the EEA. This is because higher N means also higher concentration and shorter distance between the molecules, which also affects the EEA. To confirm the role of g_0 in affecting the EEA process more precisely, we have followed the suggestion by the reviewer 3 and prepared new R6G cavity samples where the second cavity mode and third cavity mode are strongly coupled with the exciton energy, respectively (see supplementary figure 9), by tuning the thickness of the middle R6G layer but keeping the concentration of R6G, i.e. the distance between R6G, the same in the cavity. In this case, the collective coupling strength g_{eff} is the same for all the strongly coupled cavity samples, but g_0 is decreasing with increased thickness. Similarly, intensity dependent pump probe measurements were performed on these cavity samples with pump at 490nm and probe at 560 nm. Analysing the EEA results as shown in Supplementary Figure 9, we can see that the threshold of the EEA is lower for thinner cavity. Therefore, we can conclude that the larger the g_0 , the lower the EEA threshold because of the more efficient energy transfer. We have added the following text accordingly in main the manuscript at page 13.

'Here, we ascribed the exciton-photon coupling enhanced energy transfer to the individual exciton-photon coupling strength g_0 , which brings additional connectivity between R6G molecules as compared to the corresponding film samples. To confirm the role of g_0 in affecting the EEA process, two more strongly coupled R6G cavity samples where the second cavity mode and third cavity mode are strongly coupled with the exciton energy, respectively (see supplementary Figure 9), were prepared through tuning the thickness of the middle R6G layer but keeping the concentration of R6G, i.e. the distance between R6G, the same in the cavity. In this case, the collective coupling strength g_{eff} is the same for all the strongly coupled cavity samples, but g_0 is decreasing with increased

thickness. Similar strong coupling of the high-order cavity mode with exciton energy has been reported for other dye molecules⁴³. Furthermore, intensity dependent pump probe measurements were performed on these cavity samples with pump at 490nm and probe at 560 nm. Analysing the EEA results as shown in Supplementary Figure 9, we can see that the threshold of the EEA is lower for thinner cavity. Therefore, we can conclude that the larger the g_0 , the lower the EEA threshold, demonstrating the more efficient cavity-enhanced energy transfer.'

Reviewer #2 (Remarks to the Author):

The authors present a comprehensive investigation of the cavity-mediated energy transfer process in LH2 complexes by performing intensity-dependent pump-probe spectroscopy experiments in a complete series of LH2 complexes in Fabry-Pérot cavities in the strong, medium, and weak coupling regimes together with the control systems with LH2 in bare films in the same concentrations as the ones used to prepare the cavity samples. In addition, in order to study the general applicability of optical microcavities to enhance energy transfer among chromophores, the authors present a similar study as performed for LH2 on a simple dye molecule (Rhodamine 6G). The results demonstrate an enhanced exciton-exciton annihilation between LH2 complexes, which in turn demonstrate enhanced excitation energy transfer between complexes, when they are within an optical microcavity even in the weak coupling regime. The authors ascribe the enhanced energy transfer to the greater connectivity among complexes due to the resonant cavity mode.

Overall, the manuscript is clearly written, the methods are properly described, and the conclusions are well-supported by the data presented. The work presented has wide-reaching implications since it presents a design strategy to optimize artificial photosynthetic devices, such as solar cells.

In summary, I consider that this work is in line with the journal's high quality multidisciplinary research standards. I recommend this manuscript for publication in Nature Communications.

However, I do have some comments/questions that I would like the authors to address.

Comments

1) Page 4

"According to the absorbance of the corresponding bare LH2 film on glass samples (see Figure 1b), the average distance between LH2s was estimated. For the high concentration LH2 film as well as the strongly coupled LH2 cavity, the average centre-to-centre distance of LH2s is around 9 nm."

Could the authors explain in the Methods section how is this calculation performed?

Answer: The details has been added to the Methods. 'The concentration of LH2 in PVA film indicated as high, intermediate and low is 2.5 mM, 1 mM, and 0.4 mM, respectively, which are calculated based on the Beer-Lambert law, knowing the absorbance of the films, the thickness of the film around 300 nm and the molar extinction coefficient of LH2³⁷ at 860 nm of $1.67 \times 10^6 \text{ M}^{-1} \text{ cm}^{-1}$. The distance between LH2s is calculated as⁴⁷:

$$d = \frac{1.18}{\sqrt[3]{C}},$$

Where C is the concentration (mol/L) of LH2 in PVA film..'

2) Page 5

"Figure 2 (a) shows the broadband pump probe spectra of the strongly coupled LH2 cavity sample and the corresponding high concentration bare LH2 film sample at 1-2 ps, respectively."

What does "1-2 ps" mean? Does it refer to the average of the data collected between 1 and 2 ps?

Please, clarify.

Typo: "The lifetimes corresponding to the different pump intensities are plotted in Figure 3c (black curve)." Fig3c should be Fig2d.

Answer: Thanks for pointing it out. '1-2 ps' means the average over 1 to 2 ps, which has been clarified in the corresponding sentences of *'Figure 3 (a) shows the broadband pump probe spectra of the strongly coupled LH2 cavity sample and the corresponding high concentration bare LH2 film sample which are averaged over 1 to 2 ps, respectively.'* in page 6

The typo has been corrected.

3) Page 7

I think that these two sentences: "Also, the EEA process was less efficient in the low concentration LH2 film, compared with the high concentration LH2 film, in terms of the pump intensity to initiate the EEA. Since the distance between LH2s in the former case is much larger than the latter case." should be merged.

Answer: The sentences were revised as *'The EEA process was less efficient in the low-concentration LH2 film compared to the high-concentration LH2 film, since the distance between LH2s in the former case is much larger than in the latter case.'*

4) Page 10

Typo: In the sentence "Figure 4d presented the lifetimes fitted from the exponential decays . . ." The verb "presented" should be "presents" to be consistent with the rest of the text.

Answer: We thank the referee for pointing this out. Corrected.

5) Page 11

The sentence: "The middle active layer was prepared by spin coating a polymer solution which was dispersed different concentrations of the chromophores." Should not be something like: "The middle active layer was prepared by spin coating a polymer solution in which different concentrations of the chromophores were dispersed"?

Answer: This text was changed.

Comments on figures

6) Fig1. To allow for better comparison among all LH2 samples studied and to better visualize the energy levels splitting induced by the coupling to the cavity mode, it will be good to display the absorption data for the three film samples in panel (b) and the three cavity samples in panel (c) which is currently shown in figS5.

Answer: The Figure and corresponding caption have been updated.

7) Fig2a. Only the pump-probe data at 1-2 ps is shown (also in Fig3a). What is the reason for that? Could the authors show the pump-probe data for more delay times and discuss the spectral evolution of the different samples under investigation?

Answer: The reason why we only showed the pump-probe data at one delay time is to make the comparison between the strongly coupled cavity sample and the bare LH2 film sample more explicit, displaying the extra features created by the polaritons. Also, in contrast to the similar pump probe spectra between weakly coupled cavity and the film sample, it is also an evidence of the achievement of the strong exciton-photon coupling.

We have added the pump-probe data for more delay times and discussion of the spectral evolution of the different samples under investigation to supplementary Figure 2 and supplementary Note 1:

'The broadband pump probe spectra for more delay times of strongly and weakly coupled LH2 cavity samples, and high and low concentration LH2 films are plotted in Supplementary Figure 2. We can see that the pump probe spectra shapes for high concentration film, low concentration film and weakly coupled cavity sample are quite similar with obvious negative signals around 875nm which corresponds to the GSB of B850 band and positive signals around 830nm which corresponds the ESA of the B850 band. While the pump probe data of the strongly coupled LH2 cavity presented an obvious deviation from the other three samples. Here, we employed the rate-based kinetic model as proposed previously¹ to simulate the broadband pump probe spectra for all delays. We can see that the simulated model results agree well with the experimental data for all the wavelength range, which validates this model and the corresponding energy relaxation pathway in the strongly coupled cavity system after the photoexcitation of the B800 band, i.e. B800 to B850_MP, B850_MP to DS, DS to B850_LP, B850_LP to DS, DS to ground states and B850_LP to ground states.'

8) In Fig S5c I suggest to order the legend as in panel (a) for consistency.

Answer: Figure S6 in revised manuscript has been updated.

9) Figures displaying absorption and transmission data: the use of the terms "absorption" and "transmission" in the figures seems to be misplaced in some of the graphs. For instance, in Fig1b "absorption" is used whereas in Fig1d "normalized transmission" is used; but both graphs show "absorption" data.

Answer: For the cavity sample, the transmission spectrum looks quite similar to the absorption spectrum. Here we only measure the transmission of the cavity just to show that the cavity mode is resonant with the b850 band and the negligible splitting indicated the weak coupling. Also, transmission spectra of exciton-photon coupled cavity systems have widely been shown in other reports (Nat Commun, 2018, 9, 2273; J. Phys. Chem. Lett. 2021, 12, 4944–4950; Nat Commun, 2024, 15, 10529).

10) When comparing Fig4a and Fig2a in reference 17 (previous report from the authors) the graphs show equivalent data but in the current manuscript "Transmission (%)" is used while "Absorption (%)" is used in the previous report. Also, in Fig4b (current manuscript) two absorption spectra are displayed but, in the legend text, "transmission" is used for the cavity sample while "absorption" is used for the film sample, whereas in a similar graph in the previous report in Fig2b "absorption" is used for both cavity and film sample.

Obviously, an "absorption" spectrum is a "transmission" measurement but to use the term "transmission" and show "absorption" data is confusing and not correct. I suggest to always use "absorption" (as it is traditionally used in the photosynthesis research field and to ease comparison with the vast amount of data available for LH2 and as it is done with the pump-probe data displayed in units comparable with the majority of the studies conducted for LH2 outside cavities).

Answer: For bare film sample, since the reflection is negligible, the absorbance can be obtained from just measuring the transmission, using $A = -\log T$. For the cavity samples, since the reflection signal is also very strong, it is necessary to measure both transmission and reflection signals to get the absorption spectra, which was what we did in our previous work. Also, it is essential to measure the absorption signal from the cavity if precise value of the Rabi splitting of the strongly coupled system is needed as in our first report of strong coupling in LH2 cavities. Here we only strive for a semi-

quantitative demonstration of the strong coupling. Therefore, in Figure 1d, 5a and 5b in current manuscript, only transmission signals were measured for the cavity samples. Figure 1d and 5b are mostly shown to demonstrate that the cavities are in the weak coupling regime, where there is negligible or no Rabi splitting. For that, measuring transmission spectra of the cavities are enough. Figure 5a presents the angle resolved transmission spectra of R6G cavity, which is to demonstrate that the exciton-photon coupling is in the strong coupling regime. Here we observed an energy splitting of 335 meV. According to the widely used criterion for strong coupling $\hbar\Omega_R \geq (\gamma_C + \gamma_M)/2$, where $\hbar\Omega_R$ is the Rabi splitting, γ_C is the bandwidth of the cavity, and γ_M is the bandwidth of the exciton. We find that the band splitting is much larger than $(\gamma_C + \gamma_M)/2 = (150\text{meV} + 128\text{meV})/2 = 139$ meV. Even if considering some variation of the energy splitting between absorption (true Rabi splitting) and transmission signal, it is still obvious that we are in the strong coupling regime. Therefore, we only show the angle resolved transmission data in Figure 5a.

Reviewer #3 (Remarks to the Author):

The manuscript describes a time-resolved spectroscopy study of Light-Harvesting Complex 2 (LH2) containing PVA films in an optical cavity, in both the weak and strong light-matter coupling regimes. The strong light-matter coupling regime was characterized by the formation of polariton branches, which are identified by anticrossings of the cavity dispersion and the LH2 excitation maxima in the angle- and energy-resolved absorption spectrum.

By varying the intensity of the pump laser, the authors investigated how the cavity in the weak and strong coupling regimes affects the exciton-exciton annihilation (EEA) process in LH2, which is believed to involve the exchange of a photon between two excited bacteriochlorophylls. By comparing the excited-state lifetime of the LH2-cavity systems for various pump laser fluences, to the lifetime in the bare LH2 film, the authors find that the intensity threshold for EEA is significantly reduced in the cavities for both weak and strong coupling conditions. The authors interpret these findings as evidence for an enhanced energy transfer between the individual LH2 complexes in the cavities.

While enhanced energy transfer over distances exceeding the Foerster radius, has been observed under strong coupling conditions, we are not aware of reports of enhanced energy transfer also in the weak coupling limit. Also, from a theoretical perspective, enhanced energy transfer in the strong coupling regime seems (reasonably) well understood, but not for weak coupling. We therefore believe that the findings described in this manuscript will of high interest to a broad readership, and may motivate new research, both theory and experiment. We also share the author's opinion that the findings may have implications for artificial light-harvesting, as weak coupling is much easier to realize in practical applications, than strong coupling.

However, before we recommend publication, we would like the authors to clarify a few aspects, which are outlined below.

1) Our main concern is that the conclusions are based on measuring the lifetime at a single wavelength. The reason for our concern is that at the 875 nm probe, the difference spectrum, shown

in Fig. 2a seems quite congested. Is it understood and does it matter, what transitions are probed: ground state bleach (GSB), stimulated emission (SE) or excited state absorption?

Answer: The probe wavelength of 875nm corresponds to the GSB of the LP of the strongly coupled LH2 cavity sample and the GSB of the bare B850 band exciton. These are the states where EEA takes place. Kinetics corresponding to the decay of that band would show similar EEA behaviour. We did check also the pump intensity dependent kinetics around 850nm for the strongly coupled LH2 cavity sample, where the pump probe signal was the strongest. Compared with the kinetics around 875nm, as shown in the comment Figure 1, we can see that the lifetimes at the two wavelengths are the same, while the initial rising time at 850nm is a bit faster than the one at 875nm. Since the involved states share the same ground state, it is expected that the same lifetime is observed. The difference at the initial rising time between 850nm and 875nm originates from the energy relaxation pathway from MP to DS and DS to LP.

2) Furthermore, is the extent of Rabi contraction intensity dependent? If so, this would only affect the evolution of the signal at 875 nm in the strongly coupled cavity, but not in the weakly coupled cavity or in the bare film.

Answer: The magnitude of Rabi contraction is dependent on the pump intensity while, according to the Figure 6 in Musser's article (JCP 155 (2021) 154701) the shape of the contribution to the pump-probe spectra resulting from the Rabi contraction remains the same. Furthermore, the signal is linear with the pump intensity at least until the excitation concentration of 1%. Therefore, as long as the lowest two pump excitation densities where the lifetime shortening is observed are within this range, we can exclude the effect from the Rabi contraction when determining the nonlinear EEA threshold. The lowest two pump intensities ($2.8 \mu\text{J}/\text{pulse}/\text{cm}^2$ and $8.5 \mu\text{J}/\text{pulse}/\text{cm}^2$) for the strongly coupled LH2 cavity sample correspond to an LH2 excitation probability of 0.02 and 0.07, respectively. Taking into account the number of bacteriochlorophyll (BChl) a molecules per LH2 ring we obtain the excitation density of 0.07% and 0.25%, respectively. Both are well below the limit of 1%, which excludes the effect of Rabi contraction. Here is the relevant text in the revised manuscript in page 7:

'Here, we exclude that the EEA threshold could be affected by the untargeted effects that can be induced by the pump excitation, e.g. Rabi contraction, thermal expansion of the cavity, and bulk refractive index changes⁴¹. These non-polaritonic effects on the transient signals have been discussed and quantified in our previous report¹⁷. Supplementary Figure 4 shows the calculated pump probe spectrum from these non-specific photoinduced effects based on the coupled oscillator model¹⁷, which has a very different spectral shape and much weaker intensity compared to the experimentally measured spectrum. We point out that the strongest of these effects by large margin is the Rabi contraction which clearly depends on the excitation intensity. However, as reported by Musser and coworkers⁴¹ the spectral shape of the transient signal that is related to the Rabi contraction is independent of excitation intensity and the signal strength scales linearly with the excitation intensity beyond 1% of the excitation concentration. This is much larger than the lowest two pump excitation densities where the lifetime shortening was still observed in this work. Therefore we can exclude that the untargeted effects can influence the EEA threshold of the strongly coupled cavity sample.'

3) Focusing on the evolution of the transmission at a single wavelength may thus mask other processes. Therefore, would it be possible to acquire time-traces also for other wavelengths, in particular at 850 nm, where the difference signal seems strongest? Finding identical decay rates at

multiple wavelengths would reinforce the conclusion that the increased decay rate at increasing intensity is indeed due to enhanced EEA.

Answer: See also the answer to the comment 4 of the reviewer 1. The time trace around 850nm of the strongly coupled LH2 cavity sample was obtained from the broadband pump probe measurement. Compared with the time trace around 875nm as shown in comment Figure 1, we can see that the lifetimes as well as the onset of the EEA process at the two wavelengths are the same, which reinforces our conclusion of enhanced EEA by cavity coupling. The initial rising time at 850nm is a bit faster than the one at 875nm. The difference is caused by the energy relaxation pathway from MP to DS and DS to LP.

4) The identification of the EEA thresholds seems based on a single data point in Fig. 2-4 (i.e. when these last two points are on a horizontal line). Leaving one point out could alter the conclusion, therefore. Would it be possible to acquire at least one more lifetime below threshold?

Answer: Since the x axis is in log scale, the intensity difference for the lowest two points is two times. This is sufficient to identify the EEA threshold. Reducing the intensity by another factor two would significantly reduce the signal to noise ratio.

5) An interesting observation is that for the LH2 cavity systems, the slopes of the lifetime versus the logarithm of intensity are different in the cavity and in the film. For the Rhodamine-6G (R6G) cavities, these slopes are identical. Do the authors have an explanation for this.

Answer: Thank you for the question. This is an interesting point but goes beyond the current study. It will be further investigated in the future where we plan to include a few more exciton-photon coupled systems to find the factors that affect the slopes. Since the EEA process can be affected by many factors, such as the size of the molecular system or the diffusion length and mobility of exciton, the distance between molecules, the lifetime of the molecules etc., it is not simple to conclude what makes the slopes different.

6) For the R6G system, the pump laser excites the lower polariton (LP), which is delocalized over many molecules, whereas for the LH2 system, the pump laser excites a chlorophyll in the B800 ring, which is a local excitation that then transfers into the polariton manifold that is formed due to the strong coupling of the B850 ring with the cavity mode. Because, as the authors point out, the relaxation pathway after excitation into the LP is different, we do not understand why a different excitation scheme was chosen. We believe that for a comparison between R6G and LH2, both systems should be excited off-resonantly into a dark state of the cavity-molecule system. This could, for example, be achieved if a higher lying electronic or vibronic state of R6G is pumped. We are not aware of any reports of such experiments, so perhaps this is not that easy. In that case, we suggest that authors discuss in more detail the implications of the different excitation conditions before generalizing their findings.

Answer: We carried out new intensity dependent pump probe measurements on the strongly coupled R6G cavity by exciting the UP with the pump wavelength around 490nm and probe around 560 nm. These are added as a supplementary result (see supplementary Figure 8) complementing the one already shown in the manuscript where LP was excited. The EEA was analysed and compared between the strongly coupled cavity and the bare R6G film. Likewise, the EEA threshold is still lower for the cavity sample than the bare R6G film, indicating that the EEA process is independent of the

pump wavelength. We have added the following text in page 13 : *'In addition, we also evaluated the relation of EEA on pump wavelength to generalize the conclusion of cavity enhanced EEA by pumping at 490nm exciting the UP of the strongly coupled R6G cavity (see Supplementary Figure 8). Analogously, an enhanced EEA was detected for the strongly coupled R6G cavity, compared with the bare R6G film, indicating that the EEA is independent of the pump wavelength.'*

7) The authors base their conclusion of enhanced transport under weak and strong light-matter coupling conditions on a reduced excited-state lifetime. Because we think that this connection may not be immediately clear to the readers, we suggest to explain in more detail the EEA mechanism and the role of exciton transfer in that process. Perhaps a schematic illustration would also help.
Answer: This has been added in the text in page 6 and Figure 2.

'EEA is a process which involves the interaction of two excitons that can transfer energy within an array of molecules. As a consequence of the transfer, the two excitons can meet at a single molecule forming a double exciton – a higher excited molecular state which rapidly relaxes down the lowest excited state, resulting in the annihilation of one exciton. Overall, the population of excited states decreases due to the EEA. With increased excitation intensity, the probability of EEA increases leading to a faster decay of the excited states and a shorter lifetime. Thus, excitation intensity dependent lifetime measurements are widely used to characterize the exciton transfer via EEA process.'

8) The authors attribute the enhanced transfer of the excitation to the single molecule coupling (g_0), but also conclude that the number of molecules (or equivalently the number of dark states) plays an important role as well. Considering that the cavity with the highest concentration is just on the verge of strong coupling, also the cavity with the intermediate LH2 concentration must be in the weak coupling regime, yet the threshold is higher than in the cavity with the lowest concentration (Figure S5c). This would support the notion that the number of molecules is more important than the single molecule coupling. To understand the interplay between concentration, single molecule coupling and collective coupling better, would it be possible to vary the cavity thickness to be on resonance with a higher-order cavity mode, as in Bhuyan et al. (Adv. Opt. Mat. 12 (2024) 2301383)? This way, the single molecule cavity coupling can be reduced without affecting the collective coupling strength.
Answer: The number of molecules N is not the only relevant difference between strongly and weakly coupled cavities. The distance between molecules is also important. The different EEA threshold in different concentrations of bare films demonstrates the role of the distance between molecules on EEA. Therefore, a direct comparison between the three cavity samples and concluding the role of the number of molecules N is not rigorous and we have revised the text in page 10 accordingly: *'The relevant differences between the strongly and weakly coupled cavity samples are the number of LH2s inside the cavity mode volume and the distance between LH2s. Comparing the EEA process in the strongly and weakly coupled LH2 cavity samples, we can see that the pump intensity threshold to initiate EEA is higher in the weakly coupled sample, indicating that the number of LH2s inside the cavity and the distance between LH2s play a role for the EEA efficiency.'* Also, the EEA process was less efficient in the low concentration LH2 film, compared to the high concentration LH2 film, since the distance between LH2s in the former case is much larger than the latter case, verifying the significance of the distance between molecules on the EEA process.'

We appreciate your suggestion about the cavities with higher order modes and have carried out these experiments and added the following text in page 13.

'Here, we ascribed the exciton-photon coupling enhanced energy transfer to the individual exciton-photon coupling strength g_0 , which brings additional connectivity between R6G molecules as compared to the corresponding film samples. To confirm the role of g_0 in affecting the EEA process, two more strongly coupled R6G cavity samples where the second cavity mode and third cavity mode are strongly coupled with the exciton energy, respectively (see supplementary Figure 9), were prepared through tuning the thickness of the middle R6G layer but keeping the concentration of R6G, i.e. the distance between R6G, the same in the cavity. In this case, the collective coupling strength g_{eff} is the same for all the strongly coupled cavity samples, but g_0 is decreasing with increased thickness. Similar strong coupling of the high-order cavity mode with exciton energy has been reported for other dye molecules⁴³. Furthermore, intensity dependent pump probe measurements were performed on these cavity samples with pump at 490nm and probe at 560 nm. Analysing the EEA results as shown in Supplementary Figure 9, we can see that the threshold of the EEA is lower for thinner cavity. Therefore, we can conclude that the larger the g_0 , the lower the EEA threshold, demonstrating the more efficient cavity-enhanced energy transfer.'

9) We have not understood the derivation in Supplementary Note 1, from which the authors conclude that the intensity inside the cavity is the same as outside. Intuitively, at every encounter with the mirrors there is loss due to transmission, while already at the first encounter of the incoming beam there is 70% loss due to reflectivity. This is very much like a transfer matrix (TMM) calculation, and we therefore wonder if a TMM calculations with suitable optical constants would give similar results?

Answer: Although at the first encounter of the incoming beam there is 70% loss due to reflectivity, the beam will reflect back and forth many times (here is denoted as n times and is considered as infinite) inside the cavity, which is described in supplementary note 1 by a sum of a geometric progression with the reflection of the mirror as the common ratio. Compared with the TMM calculation, here we simplified the equation neglecting the phase change after the reflection, since the pump excitation wavelength is off-resonant with the cavity mode and the constructive/destructive interference effect is minimum. To make it more explicit, we add the sentence of *'Since the pump excitation wavelength is off-resonant with the cavity mode, here we simplify the equation neglecting the phase change after the reflection and the corresponding interference effect.'* to the Supplementary note 2.

10) In Figure S5c, one can see that for the lowest excitation powers, the lifetime increases with coupling strength (concentration). Are these differences due to the number of darkstates, or do these differences also depend on the overlap between the cavity spectrum and the molecular absorption spectrum, as suggested by Groenhof et al. in JPCL 10 (2019) 5476?

Answer: It is related to the number of dark states. Since the elongation of the kinetics is related to the dark states. The more the dark states, the longer the lifetime.

11) On Line 134, the authors refer to figure 3c, but according to us, this should be Figure 2d.

Answer: This has been changed.

12) On Line 264: the authors state that a lower EEA threshold was observed for the weakly coupled R6G cavity than for the strongly coupled cavity, but from the data shown in figure 4, such threshold

was not reached, as the cavity data points do not level off to a horizontal line in panels c and d.

Answer: The comparison was made between the weakly coupled R6G cavity and the corresponding low concentration film, not between weakly coupled cavity and strongly coupled cavity. As the EEA is so effective in cavity sample, we are not able to reach the plateau, but we can conclude that the threshold is lower for weakly coupled cavity sample compared to the bare film sample.

13) Perhaps the authors can mention the wavelength in the text, on Line 131? In addition, we would consider it useful to also include the pump and probe wavelengths as lines in the spectra figures.

Answer: The pump and probe wavelengths were mentioned earlier in this paragraph. Figure 3a and 4a in the revised manuscript have been updated with lines indicating the pump and probe wavelengths.

14) We have not understood how the authors computed the pump-probe spectrum from all untargeted effects (supplementary figure 3), in particular the Rabi contraction, which we believe depends on the intensity.

Answer: The way we calculated the pump-probe spectrum from all untargeted effects was described in our previous paper. The citation was inserted in the caption of Supplementary Figure 4. We have added a brief explanation as a revised supplementary Note 4.

'The pump probe spectrum from all untargeted effects are calculated as reported previously¹. In brief, we utilized the 3 by 3 coupled oscillator model as mentioned in the main text to calculate the microcavity optical properties. The input of the model included the absorption spectrum of the strongly coupled cavity sample at 30°, which was fitted using a series of Lorentzian functions. The parameters of the model are optimized by non-linear least squares method. Based on the model with optimized parameters, the transmission spectrum change with pump excitation induced all the non-specific effects, is computed as shown in Figure S4 which corresponds to a pump intensity of 8.5 $\mu\text{J}/\text{pulse}/\text{cm}^2$ or 0.25% bleaching. The non-specific polaritonic effects include the pump-induced thermal effect which leads to the thickness and refractive index change of LH2 film and the Rabi contraction which was taken into consideration by decreasing the number of molecules participating in coupling induced by the pump excitation.'

Reviewer #4 (Remarks to the Author):

We thank you and the reviewers for the careful reading of the manuscript (NCOMMS-24-49180A) and for the feedback that has helped us to significantly strengthen our work. We have revised the manuscript and added new experimental results. We offer answer to reviewers' comment and explain the changes in the manuscript. Besides, in the submitted revision, all changes are marked.

Sincerely, on behalf of the authors,

Tõnu Pullerits

First of all, we thank the referee(s) for the thorough scrutiny of our work and for recognizing the importance of our finding that “energy transfer can be enhanced in the weak coupling regime, which admittedly would open up exciting possibilities for artificial photosynthesis.” We will not repeat the whole calculation that they provide but only address the essential question:

The authors present new results of additional experiments which support that the exciton-exciton-annihilation (EEA) is enhanced in the strong coupling regime due to enhanced energy transfer. However, related to Comment 9 in our previous review report, in which we argue that the intensity, and hence absorption in the cavity may not be the same as outside of the cavity, we are not yet fully convinced that the enhanced EEA in the weak coupling regime is also due to enhanced energy transfer. Instead, based on the arguments outlined below, we now wonder if the observed effect may perhaps be explained by a Purcell-enhancement of the absorption in this regime?

Answer:

First of all, a comment about the Purcell-enhancement – the enhancement is related to the density of states of the electromagnetic waves due to the optical cavity (can be also plasmonic structure, e.g.) which leads to the enhanced spontaneous emission rate. It is not that the material properties are altered but the field density of states is changed in the cavity. A good text about the electromagnetic wave density of states can be found for example in: William L Barnes et al 2020 J. Opt. 22 073501. When it comes to the absorption then because of the cavity resonance the field in the cavity, if in resonance, can be stronger than outside because of the interference, thereby more light can be absorbed analogously to the Purcell effect in spontaneous emission. In our previous answer we pointed out that if the laser pulses are far from the cavity resonance the phase of the field can be skipped leading to a simplified expression which leads to a conclusion that the intensity inside and outside of the cavity are the same. Our experiments with LH2 well correspond to such conditions. When it comes to the R6G that the reviewers have calculated, then the excitation at 550nm (the previous experiment) is close to the cavity resonance and the phase of the field needs to be explicitly taken into account for more precise evaluation.

We have repeated our calculations following the derivation in Chapter 3 of “Quantum Electronics for Atomic Physics” by Warren Nagourney as shown in the revised Supplementary Note 2. The total light intensity inside the cavity is calculated as the sum of the light intensity

propagating towards the right and towards the left. The amplitude of the intracavity circulating field towards the right (E_{C_R}) is described as:

$$E_{C_R} = E_0 \frac{t_1}{1 - t^2 r_1 r_2 e^{-i\delta}}$$

Where E_0 is the amplitude of the incoming light field to the cavity, t_1 is the transmission amplitude coefficients of mirror 1, r_1, r_2 are the reflection coefficients of mirror 1 and 2, respectively, which are identical in this work, t is the transmission amplitude coefficients of the absorptive film inside the cavity, δ is the phase shift as light traverses a round trip inside the cavity, which is defined as:

$$\delta = 2\pi \frac{2L}{\lambda_n} + 2\phi = 2\pi \frac{2L}{\lambda/n} + 2\phi = 2\pi \frac{2nL}{\lambda} + 2\phi$$

Where L is the length of the cavity, n is the refractive index of the film inside the cavity at the excitation wavelength λ , ϕ is the phase change after one mirror reflection, which equals π here.

Then the corresponding intensity travel towards right is:

$$I_{C_R} = I_0 \left| \frac{t_1}{1 - t^2 r_1 r_2 e^{-i\delta}} \right|^2$$

$$I_{C_R} = I_0 \frac{T}{1 + t^4 r_1^2 r_2^2 - 2t^2 r_1 r_2 \cos\delta}$$

$$I_{C_R} = I_0 \frac{T}{1 + T_F^2 R^2 - 2T_F R \cos\delta}$$

Where T is the transmission intensity coefficient of the mirror with $T=t_1^2$, R is the reflection intensity coefficient of the mirror with $R=r_1^2=r_2^2=r_1 r_2$, considering mirror 1 and 2 are identical, T_F is the transmission intensity coefficient of the intracavity film with $T_F=t^2$, which can be obtained from the absorption spectrum of the corresponding bare film sample with $T_F=10^{-A}$, where A is the absorbance of the film at the excitation wavelength.

Similarly, the amplitude of the intracavity circulating field towards the left is:

$$\begin{aligned} E_{C_L} &= E_0 * t_1 * t * r_2 * e^{-i\theta} \\ &+ E_0 * t_1 * t * r_2 * t * r_1 * t * r_2 * e^{-3i\theta} \\ &+ E_0 * t_1 * t * r_2 * t * r_1 * t * r_2 * t * r_1 * t * r_2 * e^{-5i\theta} \\ &+ \dots + E_0 * t_1 * t * r_2 * e^{-i\theta} * (t * r_1 * t * r_2 * e^{-2i\theta})^n \\ E_{C_L} &= E_0 t_1 t r_2 e^{-i\theta} \sum_{n=0}^{\infty} [r_1 r_2 t^2 e^{-2i\theta}]^n \end{aligned}$$

Where θ is the phase shift as light traverses a single trip inside the cavity with $\theta = \delta/2$. Since the terms within the sum above are all less than 1, we can use the math about geometric series:

$$\sum_{n=0}^{\infty} [r_1 r_2 t^2 e^{-2i\theta}]^n = \frac{1}{1 - t^2 r_1 r_2 e^{-2i\theta}} = \frac{1}{1 - t^2 r_1 r_2 e^{-i\delta}}$$

So:

$$E_{C,L} = E_0 \frac{t_1 t r_2 e^{-i\theta}}{1 - t^2 r_1 r_2 e^{-i\delta}}$$

The corresponding intensity is:

$$I_{C,L} = I_0 \frac{t_1^2 t^2 r_2^2}{1 + t^4 r_1^2 r_2^2 - 2 * t^2 r_1 r_2 \cos\delta}$$

$$I_{C,L} = I_0 \frac{TRT_F}{1 + T_F^2 R^2 - 2T_F R \cos\delta}$$

Then the total intensity inside cavity is:

$$I_C = I_{C,R} + I_{C,L} = I_0 \frac{T + TRT_F}{1 + T_F^2 R^2 - 2T_F R \cos\delta} = I_0 \frac{T + TRT_F}{1 + T_F^2 R^2 - 2T_F R \cos(2\pi \frac{2nL}{\lambda})}$$

For the LH2 containing cavities $T = 0.3, R = 0.7, L = 300nm, n = 1.48$, and T_F of the strongly coupled LH2 cavity $T_{F_strong} = 0.91, T_F$ of the weakly coupled LH2 cavity $T_{F_weak} = 0.96$, which are obtained from the absorbances of the corresponding bare LH2 films. Thus, the intensity inside the strongly coupled cavity for pump excitation of 785nm is:

$$I_{C_LH2_strong_785nm} = 0.9I_0 .$$

The intensity inside the weakly coupled cavity for pump excitation of 800nm is:

$$I_{C_LH2_weak_800nm} = 1.1I_0 .$$

Thereout, we can conclude that the intensities inside and outside the LH2 cavities with these off-resonant pump wavelengths are roughly the same. To make our statement more precise, we have changed the text in page 6 as *'We point out that when the pump excitation wavelength is far off the cavity resonant energy, the pump intensities inside and outside the cavity are very similar due to the multiple reflections of the light in the cavity (see Supplementary Note 2).'*

Similarly, the intensities inside the R6G cavities are calculated with pump wavelength at 490 nm (new experiments) and 550 nm, respectively.

When pump wavelength is 490 nm, $T = 0.2, R = 0.8, 2nL = 540nm, T_{F_strong} = 0.84, T_{F_weak} = 0.98$, then the intensity inside the strongly coupled cavity is:

$$I_{C_R6G_strong_490nm} = 0.9I_0$$

And the intensity inside the weakly coupled cavity is:

$$I_{C_R6G_weak_490nm} = I_0$$

When pump wavelength is 550nm, $T = 0.14, R = 0.86, 2nL = 540nm, T_{F_strong} = 0.72, T_{F_weak} = 0.96$, the intensity inside the strongly coupled cavity is:

$$I_{C_R6G_strong_550nm} = 1.5I_0$$

And the intensity inside the weakly coupled cavity is:

$$I_{C_R6G_weak_550nm} = 6I_0$$

In summary, when the pump wavelength is far off resonant with the cavity mode, the light intensity inside the cavity is similar with or even smaller than the intensity outside the cavity. While when the pump wavelength is near resonant with the cavity, the intensity inside the cavity can be much larger than the intensity outside the cavity. This should be taken into consideration when evaluating the cavity induced effects.

Considering the intensity difference inside and outside the R6G containing cavities, to validate our statement of weak coupling enhanced EEA, we have performed new intensity-dependent pump probe experiments with pump excitation at 490 nm on the weakly coupled R6G and corresponding low concentration R6G film, where the intensity inside and outside the cavity are almost the same and corresponding EEA processes are analyzed as shown in Figure 5(d) and Supplementary Figure S7. In addition, we also carried out experiments with 550 nm excitation and examined EEA processes of the same weakly coupled R6G cavity and low concentration R6G film with corrected pump excitation intensity inside the cavity considering $I_{C_R6G_weak_550nm} = 6I_0$. The results are shown in the Supplementary Figure S8. From these data, we can clearly see that with both pump excitation wavelengths, the EEA thresholds are at lower intensities for the weakly coupled cavity sample compared with the corresponding film sample, demonstrating the weak coupling enhanced energy transfer.

We have updated the EEA results of the R6G cavities and films in Figure 5 with the results obtained from pump excitation at 490 nm, which is more analogous with the experimental design of the LH2 case. Corresponding figure and text changes have been marked in the main manuscript and the supplementary file.

We thank you and the reviewers for the comments and suggestions on the manuscript (NCOMMS-24-49180B) which has helped us to significantly strengthen our work. We have revised the manuscript. We offer answer to reviewers' comment and explain the changes in the manuscript. Besides, in the submitted revision, all changes are marked.

Sincerely, on behalf of the authors,

Tõnu Pullerits

First of all, we thank the referee(s) for pointing out the possible non-polaritonic effects induced by the cavity structure, which is very important and should be taken serious consideration when attributing the cavity modified photophysical or photochemical processes to the coupling between the material and cavity modes. To highlight this point, we have added the following sentences in the manuscript page 5:

'Prior to any further analysis about the impact of strong exciton-photon coupling on the EEA process, it is important to exclude any non-polaritonic effects from the cavity structure, e.g. cavity induced light field change, which has been highlighted recently by Barnes and coworkers⁴¹, where they examined cavity effects on the photoisomerization process between spiropyran and merocyanine, and found out that the photoisomerization rates were correlated with the cavity induced absorption change. Here, we calculated the excitation intensity inside the cavity as compared with the excitation intensity outside the cavity which were employed for the bare film samples (see Supplementary Note 2). The results show that when the pump excitation wavelength is far off the cavity resonant energy, the pump intensities inside and outside the cavity are very similar, ruling out the above mentioned non-polaritonic effects.'

As for the deviation (quite minor) between the textbook-based method we are using to calculate the intensity inside the cavity and what the reviewers 3 and 4 have obtained by using downloaded code, we agree with the reviewers that it could be due to the different values of the parameters used, e.g. the mirror reflection/transmission and/or film transmission. All the parameters we are using are based on the measured experimental results. The reviewers use values of the materials' optical constants from other literature which leads to the reflection and transmission of the mirror and the organic film in the code what they are using. For the R6G cavity under weak coupling conditions, the light intensity for excitation at $\lambda = 550$ nm is enhanced by a factor of 6, and we scaled the intensity on the x-axis accordingly by a factor of 6. For the same cavity, we also calculated the transmission intensity and reflection intensity of the 550 nm excitation based on the same methodology we are using to derive the circulating field intensity inside the cavity, where $I_{T,550nm} = 0.46 I_0$ and $I_{R,550nm} = 0.3 I_0$. Thus the absorbed light intensity $I_{A,550nm} = 1 - I_{T,550nm} - I_{R,550nm} = 0.24 I_0$, which equals 6 times of the absorption of the film (4%) and is consistent with the light intensity inside cavity we calculated.

We agree that the refractive index depends on the absorbance (Kramers-Kronig relation) and thereby on the concentration of the films. However, in our experiments the cavity thickness was always chosen so that the cavity is in resonance with the R6G excitation energy of 540 nm. Therefore, we are using $2nL = 540$ nm when calculating the intensity inside the cavity for both strongly and weakly

coupled cavity. The distance L between the two mirrors is different for the strongly and weakly coupled cavity to achieve the same cavity resonance frequency in different cases. Here, we add the sentence '*The cavity modes of all the cavity samples are tuned to be resonant with the R6G exciton energy*' in the Methods part.

As for the data shown in Figure 5d and S8f of the EEA process for the weakly coupled R6G cavity sample and low concentration film, indeed, they were measured in a new batch of samples as the reviewers suggest. Since the R6G concentration in the low concentration film and weakly coupled cavity is quite low, it may vary a bit for different batch of samples, which will affect the lifetime of the film and cavity. The important thing is that the middle R6G-PVA layer in the weakly coupled cavity sample and the corresponding low concentration film are prepared using the same batch of dye solution. Also, we can see that the lifetime of the film is always longer than the corresponding weakly coupled cavity sample, which is consistent for different batches of samples.

The authors have addressed all but one of our concerns, which we elaborate on below.

The authors present new results of additional experiments which support that the exciton-exciton-annihilation (EEA) is enhanced in the strong coupling regime due to enhanced energy transfer. However, related to Comment 9 in our previous review report, in which we argue that the intensity, and hence absorption in the cavity may not be the same as outside of the cavity, we are not yet fully convinced that the enhanced EEA in the weak coupling regime is also due to enhanced energy transfer. Instead, based on the arguments outlined below, we now wonder if the observed effect may perhaps be explained by a Purcell-enhancement of the absorption in this regime?

To verify whether the overall absorption is affected by the cavity, we had suggested to complement the author’s calculation in Supplementary Note 2 with a Transfer Matrix Method calculation of the absorption. Because this was not done, we decided to perform such TMM calculations ourselves, using the python code developed by Steven Byrnes (<https://github.com/sbyrnes321/tmm>). We only did this for a Rhodamine-6G/PVA film in the silver cavity, as the complex dielectric functions are available for both this molecule [*Opt. Mat. Expr.* **12** (2022) 2855] and the silver [*Appl. Optics* **37** (1998) 5271].

In Figure A below, we show the calculated absorption spectrum of a 121 nm thick bare film. We first tried with a 130 nm film thickness, as in the experiments, but 121 nm gave better agreement. In Figure B, we show the calculated angular-resolved transmission, which we consider sufficiently similar to the transmission in panel a of Figure 5 in the manuscript. In Figure C we show the calculated transmission at 18° , which we also consider sufficiently similar to the experimental spectra shown in Figure S9a.

Figure A: absorption of high concentration Rhodamine-6G / PVA film.

Figure B: Angular-resolved transmission spectra of strongly coupled R6G containing microcavity. Note that the wavelength axis is in the opposite direction as compared to Figure 5(a).

Figure C: transmission spectrum of strongly coupled R6G containing microcavity at 18° .

Thus, having established that the TMM model can faithfully reproduce the experimental data, we then reduced the concentration of the Rhodamine-6G by a factor of ten to mimic the weakly coupled cavity system. In Figures D and E, we compare (on same scale) the absorption of the bare Rhodamine-6G/PVA films and the absorption of the same Rhodamine-6G/PVA film sandwiched between the 30 nm silver cavity mirrors. The absorption was thus computed for the active layer only, without contributions from mirror absorption, reflection or transmission.

Figure D: absorption of low concentration R6G film.

Figure E: absorption of low concentration R6G film inside cavity.

The ratio between the absorption of the Rhodamine-6G film inside and outside of the cavity at the 550 nm pump wavelength is about four, suggesting that when the material is embedded between the mirrors, four times more light gets absorbed. Therefore, to make a proper comparison between the cavity and the film, the film should have been pumped with a four times lower intensity. Because we do not have access to the raw data, we could not test if with such correction factor, the two lines shown in Figure 5(d) would become the same. If that would indeed be the case, then the higher EEA rate in the weak coupling regime could simply be due to the four-fold higher absorption cross-section within the cavity as compared to outside, and not due to enhanced energy transfer. At least for Rhodamine-6G. It could perhaps also explain why the two lines are parallel in Figure 5(b).

We may be wrong here, but because one of the main findings in the manuscript is that energy transfer can be enhanced in the weak coupling regime, which admittedly would open up exciting possibilities for artificial photosynthesis, we believe our concern should be addressed before the manuscript can be accepted for publication.

While our analysis was restricted to the Rhodamine-6G experiment for which no threshold was found under weak coupling conditions, we suggest to also run TMM calculations for the LH2. In particular, we would be interested to see if also for LH2, there could be an increase in absorption under weak coupling, and if correcting for such potential increase would align the black and red data points in Figure 4(b).

To address our concern that in the weak coupling regime, the cavity can enhance the absorption, the authors have performed their own calculations of the field intensity. In qualitative agreement with our TMM calculations, they find with their model that for the R6G cavity under weak coupling conditions, the light intensity for excitation at $\lambda = 550$ nm is enhanced by a factor of 6, and have accordingly corrected their data for that enhancement. However, it was not clear how this correction was applied, actually. Was it a scaling by factor 6 of the intensity on the x-axis?

More importantly, comparing the data points for pumping the R6G cavity at 550 nm in the previous version (Figure 5d in 530827_1_art_file_10180491_sq32n4_convrt.pdf) with the data points in the current version (Figure S8f), we see that the correction also affects the data measured for the bare film (i.e., outside of cavity). For example, in Figure 5d of the previous version, the lifetime at the lowest pump intensity for the film was close to 1000 ps (the plateau), whereas in the new version it is below 900 ps. Furthermore, the threshold for the bare film is now around $70 \mu\text{J}/\text{pulse}/\text{cm}^2$, but was around $35 \mu\text{J}/\text{pulse}/\text{cm}^2$ before. Likewise, the maximum lifetime obtained at the lowest fluence for the cavity sample was above 900 ps in the previous version, versus around 650 ps now. We do not understand how correcting the pump intensity for the enhanced absorption at $\lambda = 550$ nm can affect the lifetimes, or the data for the film outside of the cavity, for which no correction is required. Perhaps, a completely new set of samples was created and the measurements with excitation at 550 nm were repeated? That, however, was not stated, and would furthermore imply that differences between sets of cavity (and film) samples pumped at same wavelength are actually larger than between the cavity and film. These discrepancies should be clarified.

The authors have also performed a new set of experiments, in which the R6G cavity and film were pumped at $\lambda = 490$ nm, with the motivation that at this wavelength there is no absorption enhancement under neither weak nor strong coupling conditions according to their model. However, in our TMM calculations, we find that in the weak coupling regime, there is still a two-fold enhanced intensity at $\lambda = 490$ nm, which approximately matches the threshold being two times lower in the cavity than in the bare film (Figure 5d, from $\sim 45 \mu\text{J}/\text{pulse}/\text{cm}^2$ to $\sim 90 \mu\text{J}/\text{pulse}/\text{cm}^2$).

Although we cannot judge which of the two models is better, or correct (if any), in particular because of the dependence on parameters (permittivity for metals and dye layer in TMM, thicknesses, concentration (oscillator strength), etc.), one possible source of discrepancy may be the value of the refractive index (n) that was used in the models. We did not understand what value for the refractive index the authors used in their model? In their response (and SI), they state that $2nL = 540$ nm, both for pumping at $\lambda = 490$ nm and at $\lambda = 550$ nm. Since the film thickness was kept at $L = 150$ nm (Method, page 15), we infer therefore that a refractive index of $n = 1.8$ was used in their model, for both strong and weak coupling? Is this correct? If so, wouldn't the refractive index depend on concentration, and hence be lower for the weakly coupled cavity? Because the intensity depends critically on the refractive index used in the model(s) (see the figure below, which was calculated with the highlighted expression in the author's response), a (small) difference in n might lead to a large difference

in the calculated intensity enhancement. It would be good, therefore, if the authors could provide the exact values of n for both the strongly and weakly coupled cavity systems.

Figure: Field intensity inside the cavity (I) with respect to the field intensity outside the cavity (I_0) as a function of the refractive index (n) for a strongly (SC, blue) and weakly coupled (WC, red) cavity obtained with the expression of the authors with the following parameters: $\lambda = 490$ nm, $T = 0.2$, $R = 0.8$, $T_{F_strong} = 0.84$, $T_{F_weak} = 0.98$, $L = 150$ nm. The dashed line corresponds to the refractive index $n = 1.8$, where there is no large enhancement.

Because we did not model the LH2 system, all our comments pertain to the R6G system. Nevertheless, because the discussion about the refractive index may also apply to LH2, it would be good to share how the value of the refractive index, $n = 1.48$, of LH2 in PVA was estimated and if n remains the same at $\lambda = 785$ nm for the strongly coupled cavity and $\lambda = 800$ nm for the weakly coupled cavity.

Summarizing, in light of the arguments above and the discrepancies between the data for the R6G cavity pumped at 550 nm in the current and previous versions of the manuscript, we are not yet fully convinced that the reduced threshold for EEA in the weak coupling regime is due to enhanced exciton transfer rather than enhanced absorption, at least for R6G. Disentangling trivial (enhanced absorption due to higher field intensity in the cavity) and non-trivial (enhanced energy transfer via the cavity modes) effects of cavities is highly relevant, as attributing an observation to the wrong effect can easily cause confusion or controversy. As an example, we refer to a recent work by W. Barnes and co-workers, who could attribute cavity-induced changes in spiropyran ring opening [*Angew. Chem. Int. Ed.* **2012**, 51, 1592], to changes in absorption, rather than polaritonic effects [*Adv. Mater.*, **2024**, 36, 7, 2309393]. We hope such controversy can be avoided here. To assist the authors in addressing our concerns about the differences between the models, we have uploaded the jupyter notebook with our calculations, along with our report. The notebook is a bit messy, but we hope that it helps.

```

{
  "cells": [
    {
      "cell_type": "markdown",
      "id": "ad3c7fe2-4244-4770-93dc-6465d413ddb3",
      "metadata": {},
      "source": [
        "Intallation of the tmm package may be needed. Here, we use pip to do
that. This step can be skipped if tmm is available."
      ]
    },
    {
      "cell_type": "code",
      "execution_count": null,
      "id": "87ba64d0-f66b-46c2-ad0d-2211a574817a",
      "metadata": {},
      "outputs": [],
      "source": [
        "!pip3 install tmm"
      ]
    },
    {
      "cell_type": "code",
      "execution_count": null,
      "id": "2e2c3728-6394-4ed6-b837-8e2cc7b960bc",
      "metadata": {},
      "outputs": [],
      "source": [
        "# some stuff \n",
        "from __future__ import division, print_function, absolute_import\n",
        "\n",
        "\n",
        "\n",
        "import os\n",
        "from array import array\n",
        "\n",
        "\n",
        "from tmm import (coh_tmm, coh_tmm_reverse, unpolarized_RT, ellips,\n",
        "                 position_resolved, find_in_structure_with_inf,\n",
        "absorp_in_each_layer)\n",
        "\n",
        "\n",
        "from numpy import pi, linspace, inf, array\n",
        "from scipy.interpolate import interp1d\n",
        "import matplotlib.pyplot as plt\n",
        "%matplotlib inline\n",
        "\n",
        "try:\n",
        "    import colorpy.illuminants\n",
        "    import colorpy.colormodels\n",
        "    from tmm import color\n",
        "    colors_were_imported = True\n",
        "except ImportError:\n",
        "    # without colorpy, you can't run sample5(), but everything else is
fine.\n",
        "    colors_were_imported = False\n",
        "degree = pi/180\n",
        "\n",
        "Hz2eV=4.13566553853599E-15\n",
        "c=299792458"
      ]
    },
    {
      "cell_type": "code",
      "execution_count": null,
      "id": "370a27be-4586-4fc7-9ca6-d8a6760e857c",
      "metadata": {},
      "outputs": [],
      "source": [
        "# -*- coding: utf-8 -*-\n",
        "# Author: Mikhail Polyanskiy\n",
        "# Last modified: 2017-04-02\n",
        "# Original data: RakiÄ† et al. 1998,\n",
https://doi.org/10.1364/AO.37.005271\n",
        "\n",
        "import numpy as np\n",
        "import matplotlib.pyplot as plt\n",

```

```

"from scipy.special import wofz as w\n",
"iE = np.pi\n",
"\n",
"# Brendel-Bormann (BB) model parameters\n",
"i%p = 9.01 #eV\n",
"f0 = 0.821\n",
"i^0 = 0.049 #eV\n",
"\n",
"f1 = 0.050\n",
"i^1 = 0.189 #eV\n",
"i%1 = 2.025 #eV\n",
"if1 = 1.894 #eV\n",
"\n",
"f2 = 0.133\n",
"i^2 = 0.067 #eV\n",
"i%2 = 5.185 #eV\n",
"if2 = 0.665 #eV\n",
"\n",
"f3 = 0.051\n",
"i^3 = 0.019 #eV\n",
"i%3 = 4.343 #eV\n",
"if3 = 0.189 #eV\n",
"\n",
"f4 = 0.467\n",
"i^4 = 0.117 #eV\n",
"i%4 = 9.809 #eV\n",
"if4 = 1.170 #eV\n",
"\n",
"f5 = 4.000\n",
"i^5 = 0.052 #eV\n",
"i%5 = 18.56 #eV\n",
"if5 = 0.516 #eV\n",
"\n",
"iOp = f0**.5 * i%p #eV\n",
"\n",
"def BB(i%): #i%: eV\n",
"    iμ = 1-iOp**2/(i%*(i%+1j*i^0))\n",
"\n",
"    i± = (i%**2+1j*i%*i^1)**.5\n",
"    za = (i±-i%1)/(2**.5*if1)\n",
"    zb = (i±+i%1)/(2**.5*if1)\n",
"    iμ += 1j*iE**.5*f1*i%p**2 / (2**1.5*i±*if1) * (w(za)+w(zb))
#i+1\n",
"    \n",
"    i± = (i%**2+1j*i%*i^2)**.5\n",
"    za = (i±-i%2)/(2**.5*if2)\n",
"    zb = (i±+i%2)/(2**.5*if2)\n",
"    iμ += 1j*iE**.5*f2*i%p**2 / (2**1.5*i±*if2) * (w(za)+w(zb))
#i+2\n",
"    \n",
"    i± = (i%**2+1j*i%*i^3)**.5\n",
"    za = (i±-i%3)/(2**.5*if3)\n",
"    zb = (i±+i%3)/(2**.5*if3)\n",
"    iμ += 1j*iE**.5*f3*i%p**2 / (2**1.5*i±*if3) * (w(za)+w(zb))
#i+3\n",
"    \n",
"    i± = (i%**2+1j*i%*i^4)**.5\n",
"    za = (i±-i%4)/(2**.5*if4)\n",
"    zb = (i±+i%4)/(2**.5*if4)\n",
"    iμ += 1j*iE**.5*f4*i%p**2 / (2**1.5*i±*if4) * (w(za)+w(zb))
#i+4\n",
"    \n",
"    i± = (i%**2+1j*i%*i^5)**.5\n",
"    za = (i±-i%5)/(2**.5*if5)\n",
"    zb = (i±+i%5)/(2**.5*if5)\n",
"    iμ += 1j*iE**.5*f5*i%p**2 / (2**1.5*i±*if5) * (w(za)+w(zb)) #i+5
\n",
"    \n",
"    return iμ\n",
"    \n",
"ev_min=0.1\n",
"ev_max=5\n",
"npoints=200\n",
"eV = np.logspace(np.log10(ev_min), np.log10(ev_max), npoints)\n",
"i%_m = 4.13566733e-1*2.99792458/eV\n",
"iμ = BB(eV)\n",
"n = (iμ**.5).real\n",

```

```

    "k = (Îµ**0.5).imag\n",
    "\n",
    "Ag_nk_data=np.stack((1240/eV,Îµ**0.5),axis=1)\n",
    "Ag_nk_fn = interp1d(Ag_nk_data[:,0].real,\n",
    "                    Ag_nk_data[:,1], kind='quadratic')\n"
]
},
{
    "cell_type": "code",
    "execution_count": null,
    "id": "35f5eedf-efd4-47ce-928f-751fc143f0f2",
    "metadata": {},
    "outputs": [],
    "source": [
        "# fit parameters for Rh6G at high concentration, for reaching SC
        regime\n",
        "f1 = 0.032\n",
        "f2 = 0.0129\n",
        "gamma1 = 0.1497\n",
        "gamma2 = 0.1633\n",
        "omega01 = 2.2946\n",
        "omega02 = 2.4468\n",
        "alpha1 = 1.4171\n",
        "alpha2 = 0.8727\n",
        "# oscillator strength, tuned to match experiment\n",
        "f=0.5\n",
        "f1*=f\n",
        "f2*=f\n",
        "\n",
        "# modified Lorentz oscillator model for R6G. Parameters retrieved
        from Optical Materials Express Vol. 12, Issue 7, pp. 2855-2869 (2022)\n",
        "\n",
        "def gammaf1(Î): #Î: eV\n",
        "    temp = gamma1*np.exp(-alpha1*((Î-omega01)/gamma1)**2)\n",
        "    return temp\n",
        "\n",
        "def gammaf2(Î): #Î: eV\n",
        "    temp = gamma2*np.exp(-alpha2*((Î-omega02)/gamma2)**2)\n",
        "    return temp\n",
        "\n",
        "def eps(Î): #Î: eV\n",
        "    realpart = 2.25+f1*omega01**2*(omega01**2 - Î**2)/((omega01**2 -
        Î**2)**2 + (gammaf1(Î)**2)*Î**2)+f2*omega02**2*(omega02**2 -
        Î**2)/((omega02**2 - Î**2)**2 + (gammaf2(Î)**2)*Î**2)\n",
        "    imaginarypart = f1*omega01**2*(gammaf1(Î)*Î)/((omega01**2 -
        Î**2)**2 +
        (gammaf1(Î)**2)*Î**2)+f2*omega02**2*(gammaf2(Î)*Î)/((omega02**2 -
        Î**2)**2 + (gammaf2(Î)**2)*Î**2)\n",
        "    eps = realpart+1j*imaginarypart\n",
        "    return eps\n",
        "\n",
        "ev_min=0.0001\n",
        "ev_max=8\n",
        "npoints=2000\n",
        "eV = np.logspace(np.log10(ev_min), np.log10(ev_max), npoints)\n",
        "Î_m = 4.13566733e-1*2.99792458/eV\n",
        "\n",
        "Îµ = eps(eV)\n",
        "n = (Îµ**0.5).real\n",
        "k = (Îµ**0.5).imag\n",
        "Dye_nk_data=np.stack((1240/eV,Îµ**0.5),axis=1)\n",
        "Dye_nk_fn = interp1d(Dye_nk_data[:,0].real,\n",
        "                    Dye_nk_data[:,1], kind='quadratic')\n"
    ]
},
{
    "cell_type": "code",
    "execution_count": null,
    "id": "7f25511e-0821-4b9a-bb07-61fbdfcfe433",
    "metadata": {},
    "outputs": [],
    "source": [
        "d_list = [inf,121,inf] #in nm\n",
        "lambda_list = linspace(300,800,500) #in nm\n",
        "abs_R6G_per_layer = []\n",
        "theta=0\n",
        "for lambda_vac in lambda_list:\n",

```

```

    n_list = [1, Dye_nk_fn(lambda_vac), 1]\n",
    \n",
    ]
abs_R6G_per_layer.append(absorp_in_each_layer(coh_tmm('s',n_list,d_list,theta*degree,lambda_vac)))
"
]
},
{
"cell_type": "code",
"execution_count": null,
"id": "c17335bf-43d7-451b-be10-a5f473276383",
"metadata": {},
"outputs": [],
"source": [
"abs_R6G = []\n",
"for i in range(0, 500,1):\n",
"    abs_R6G.append(abs_R6G_per_layer[i][1])\n",
"\n",
"plt.figure(1)\n",
"    \n",
"plt.plot(c/(lambda_list*1e-09)*Hz2eV,abs_R6G,color='orange') \n",
"plt.xlabel('energy (eV)')\n",
"plt.ylabel('absorption')\n",
"plt.savefig('R6G_absorption.png')\n",
"\n",
"plt.figure(2)\n",
"    \n",
"plt.plot(lambda_list,abs_R6G,color='orange') \n",
"plt.xlabel('energy (eV)')\n",
"plt.ylabel('absorption')\n",
"plt.savefig('R6G_absorption_nm.png')"
]
},
{
"cell_type": "code",
"execution_count": null,
"id": "8c844289-7a60-4cee-b429-140fb5c864fd",
"metadata": {},
"outputs": [],
"source": [
"# Cavity with silver mirrors, dispersion plot\n",
"\n",
"d_list = [inf,30,121,30,inf] #in nm\n",
"lambda_list = linspace(580,420,400) #in nm\n",
"T_list = []\n",
"\n",
"\n",
"theta_list = linspace(0.0,50,400)\n",
"for theta in theta_list:\n",
"    \n",
"    for lambda_vac in lambda_list:\n",
"        \n",
"            n_list = [1, Ag_nk_fn(lambda_vac), Dye_nk_fn(lambda_vac)
,Ag_nk_fn(lambda_vac), 1]\n",
"            T_list.append(coh_tmm('s',n_list,d_list,theta*degree,lambda_vac) ['T'])\n",
"            \n",
"            \n",
"            data=array(T_list)\n",
"            datanew = data.reshape(400,400)\n",
"            Y, X = np.meshgrid(lambda_list,theta_list )\n",
"            fig,ax=plt.subplots(1,1)\n",
"            cp = ax.contourf(X, Y, datanew)\n",
"            fig.colorbar(cp) # Add a colorbar to a plot\n",
"            \n",
"            ax.set_xlabel('angle (degree)')\n",
"            ax.set_ylabel('Wavelength (nm)')\n",
"            plt.show() "
]
},
{
"cell_type": "code",
"execution_count": null,
"id": "9d584948-a810-48f6-a5f9-3a0d349ad11b",
"metadata": {},
"outputs": [],
"source": [
"# Cavity with silver mirrors, some 1D spectra to compare to

```

```

experiments\n",
"\n",
"d_list = [inf,30,122,30,inf] #in nm\n",
"lambda_list = linspace(450,800,500) #in nm\n",
"T_list = []\n",
"abs_R6G_cav_per_layer = []\n",
"\n",
"theta = 18 \n",
" \n",
"for lambda_vac in lambda_list:\n",
" \n",
" n_list = [1, Ag_nk_fn(lambda_vac), Dye_nk_fn(lambda_vac)
,Ag_nk_fn(lambda_vac), 1]\n",
"
T_list.append(coh_tmm('s',n_list,d_list,theta*degree,lambda_vac) ['T'])\n",
"
abs_R6G_cav_per_layer.append(absorp_in_each_layer(coh_tmm('s',n_list,d_list,theta*degree,lambda_vac)))\n"
,
"\n",
"\n",
"plt.figure(1) \n",
"plt.plot(lambda_list,T_list,color='red') \n",
"plt.xlabel('wavelength (nm)')\n",
"plt.ylabel('Transmission')\n",
"plt.savefig('R6G_cavity_transmission_nm.png')\n",
"\n",
"plt.figure(2)\n",
"plt.plot(c/(lambda_list*1e-09)*Hz2eV,T_list,color='black') \n",
"plt.xlabel('energy (eV)')\n",
"\n",
"\n",
"abs_R6G_cav = []\n",
"for i in range(0, 500,1):\n",
" abs_R6G_cav.append(abs_R6G_cav_per_layer[i][2])\n",
"plt.figure(3)\n",
" \n",
"plt.plot(lambda_list,abs_R6G_cav,color='red') \n",
"plt.xlabel('energy (eV)')\n",
"plt.ylabel('absorption')\n",
"plt.savefig('R6G_layer_absorption_nm.png')\n",
"\n"
]
},
{
"cell_type": "code",
"execution_count": null,
"id": "84441812-5452-4fd3-93e0-5a588e96590a",
"metadata": {},
"outputs": [],
"source": [
"# Cavity with silver mirrors, smaller concentration of R6G,
corresponding to weak coupling regime\n",
"# fit parameters for Rh6G\n",
"f1 = 0.032\n",
"f2 = 0.0129\n",
"gamma1 = 0.1497\n",
"gamma2 = 0.1633\n",
"omega01 = 2.2946\n",
"omega02 = 2.4468\n",
"alpha1 = 1.4171\n",
"alpha2 = 0.8727\n",
"# overall oscillator strenght, factor 10 smaller than in the SC
situation. \n",
"f=0.05\n",
"f1*=f\n",
"f2*=f\n",
"\n",
"# modifield Lorentz oscillator model\n",
"\n",
"def gammaf1(i%): #i%: eV\n",
" temp = gamma1*np.exp(-alpha1*((i%-omega01)/gamma1)**2)\n",
" return temp\n",
"\n",
"def gammaf2(i%): #i%: eV\n",
" temp = gamma2*np.exp(-alpha2*((i%-omega02)/gamma2)**2)\n",
" return temp\n",
"\n",
"def eps(i%): #i%: eV\n",

```

```

"    realpart = 2.25+f1*omega01**2*(omega01**2 - i**2)/((omega01**2 -
i**2)**2 + (gammaf1(i)**2)*i**2)+f2*omega02**2*(omega02**2 -
i**2)/((omega02**2 - i**2)**2 + (gammaf2(i)**2)*i**2)\n",
"    imaginarypart = f1*omega01**2*(gammaf1(i)*i)/((omega01**2 -
i**2)**2 +
(gammaf1(i)**2)*i**2)+f2*omega02**2*(gammaf2(i)*i)/((omega02**2 -
i**2)**2 + (gammaf2(i)**2)*i**2)\n",
"    eps = realpart+1j*imaginarypart\n",
"    return eps\n",
"ev_min=0.0001\n",
"ev_max=8\n",
"npoints=2000\n",
"eV = np.logspace(np.log10(ev_min), np.log10(ev_max), npoints)\n",
"i_m = 4.13566733e-1*2.99792458/eV\n",
"\n",
"i_u = eps(eV)\n",
"n = (i_u**.5).real\n",
"k = (i_u**.5).imag\n",
"Dye_nk_data=np.stack((1240/eV, i_u*.5), axis=1)\n",
"Dye_nk_fn = interp1d(Dye_nk_data[:,0].real, \n",
"                    Dye_nk_data[:,1], kind='quadratic')\n",
"\n",
"# calculate and plot spectra\n",
"d_list = [inf,121,inf] #in nm\n",
"lambda_list = linspace(300,800,500) #in nm\n",
"abs_R6G_per_layer = []\n",
"theta=0\n",
"for lambda_vac in lambda_list:\n",
"    n_list = [1, Dye_nk_fn(lambda_vac), 1]\n",
"    \n",
abs_R6G_per_layer.append(absorp_in_each_layer(coh_tmm('s',n_list,d_list,theta*degree,lambda_vac))\n",
',
"\n",
"abs_R6G = []\n",
"for i in range(0, 500,1):\n",
"    abs_R6G.append(abs_R6G_per_layer[i][1])\n",
"\n",
"plt.figure(1)\n",
"    \n",
"plt.plot(c/(lambda_list*1e-09)*Hz2eV,abs_R6G,color='orange') \n",
"plt.xlabel('energy (eV)')\n",
"plt.ylabel('absorption')\n",
"\n",
"plt.savefig('low_R6G_layer_absorption_nm.png')\n",
"plt.figure(2)\n",
"    \n",
"plt.plot(lambda_list,abs_R6G,color='orange') \n",
"plt.xlabel('wavelength (nm)')\n",
"plt.ylabel('absorption')\n",
"plt.savefig('low_R6G_absorption_nm.png')\n",
"\n",
"\n",
"d_list = [inf,30,121,30,inf] #in nm\n",
"lambda_list = linspace(300,800,500) #in nm\n",
"T_list = []\n",
"abs_R6G_cav_per_layer = []\n",
"\n",
"theta = 18 \n",
"    \n",
"for lambda_vac in lambda_list:\n",
"    \n",
"    n_list = [1, Ag_nk_fn(lambda_vac), Dye_nk_fn(lambda_vac)
,Ag_nk_fn(lambda_vac), 1]\n",
"
T_list.append(coh_tmm('s',n_list,d_list,theta*degree,lambda_vac) ['T'])\n",
abs_R6G_cav_per_layer.append(absorp_in_each_layer(coh_tmm('s',n_list,d_list,theta*degree,lambda_vac))\n",
',
"\n",
"\n",
"plt.figure(3) \n",
"plt.plot(lambda_list,T_list,color='red') \n",
"plt.xlabel('wavelength (nm)')\n",
"plt.ylabel('Transmission')\n",
"\n",
"plt.figure(4)\n",

```

```

"plt.plot(c/(lambda_list*1e-09)*Hz2eV,T_list,color='black') \n",
"plt.xlabel('energy (eV)')\n",
"\n",
"abs_R6G_cav = []\n",
"for i in range(0, 500,1):\n",
"    abs_R6G_cav.append(abs_R6G_cav_per_layer[i][2])\n",
"plt.figure(5)\n",
"    \n",
"plt.plot(lambda_list,abs_R6G_cav,color='red') \n",
"plt.xlabel('wavelength (nm)')\n",
"plt.ylabel('absorption')\n",
"plt.savefig('low_R6G_layer_absorption_nm.png')"
]
},
{
"cell_type": "code",
"execution_count": null,
"id": "e8dfbfff2-6c66-415d-81cc-e32bbff93614",
"metadata": {},
"outputs": [],
"source": [
"#ratio of absorption between cavity and film at 490 nm\n",
"abs_R6G_cav[190]/abs_R6G[190]"
]
},
{
"cell_type": "code",
"execution_count": null,
"id": "fd226cbf-fa59-46f4-8978-b7d485de1198",
"metadata": {},
"outputs": [],
"source": [
"# ration of absorption between cavity and film at 550 nm\n",
"abs_R6G_cav[250]/abs_R6G[250]"
]
},
{
"cell_type": "code",
"execution_count": null,
"id": "49b522fc-fd98-4d0c-8700-6d396221e4ee",
"metadata": {},
"outputs": [],
"source": []
}
],
"metadata": {
"kernel_spec": {
"display_name": "Python 3 (ipykernel)",
"language": "python",
"name": "python3"
},
"language_info": {
"codemirror_mode": {
"name": "ipython",
"version": 3
},
"file_extension": ".py",
"mimetype": "text/x-python",
"name": "python",
"nbconvert_exporter": "python",
"pygments_lexer": "ipython3",
"version": "3.9.6"
}
},
"nbformat": 4,
"nbformat_minor": 5
}

```